# What factors are associated with reporting lacking interest in sex and how do these vary by gender? Findings from the third British national survey of sexual attitudes and lifestyles

Cynthia A Graham,[1] Catherine H Mercer,[2] Clare Tanton,[2] Kyle G Jones,[2] Anne M Johnson,[2] Kaye Wellings,[3] Kirstin R Mitchell[4]

[1]Department of Psychology, Centre for Sexual Health Research, University of Southampton, Southampton, UK
[2]Centre for Sexual Health and HIV Research, Research Department of Infection & Population Health, University College London, London, UK
[3]Centre for Sexual and Reproductive Health Research, Department of Social and Environmental Health Research, London School of Hygiene and Tropical Medicine, London, UK
[4]MRC/CSO Social and Public Health Sciences Unit, University of Glasgow, London, UK

**Correspondence to**
Dr Cynthia A Graham;
C.A.Graham@soton.ac.uk

## ABSTRACT

**Objectives** To investigate factors associated with reporting lacking interest in sex and how these vary by gender.

**Setting** British general population.

**Design** Complex survey analyses of data collected for a cross-sectional probability sample survey, undertaken 2010–2012, specifically logistic regression to calculate age-adjusted OR (AOR) to identify associated factors.

**Participants** 4839 men and 6669 women aged 16–74 years who reported ≥1 sexual partner (opposite-sex or same-sex) in the past year for the third National Survey of Sexual Attitudes and Lifestyles (Natsal-3).

**Main outcome measure** Lacking interest in sex for ≥3 months in the past year.

**Results** Overall, 15.0% (13.9–16.2) of men and 34.2% (32.8–35.5) of women reported lacking interest in sex. This was associated with age and physical and mental health for both men and women, including self-reported general health and current depression. Lacking interest in sex was more prevalent among men and women reporting sexually transmitted infection diagnoses (ever), non-volitional sex (ever) and holding sexual attitudes related to normative expectations about sex. Some gender similarities in associated relationship and family-related factors were evident, including partner having had sexual difficulties in the last year (men: AOR 1.41 (1.07–1.86); women: AOR 1.60 (1.32–1.94)), not feeling emotionally close to partner during sex (men: 3.74 (1.76–7.93); women: 4.80 (2.99–7.69) and ease of talking about sex (men: 1.53 (1.23–1.90);women: 2.06 (1.77–2.39)). Among women only, lack of interest in sex was higher among those in a relationship of >1 year in duration and those not sharing the same level of interest (4.57 (3.87–5.38)) or preferences (2.91 (2.22–3.83)) with a partner.

**Conclusions** Both gender similarities and differences were found in factors associated with lacking interest in sex, with the most marked differences in relation to some relationship variables. Findings highlight the need to assess, and if appropriate, treat lacking interest in sex in a holistic and relationship-specific way.

### Strengths and limitations of this study

► This study used nationally representative data to investigate factors associated with reporting lacking interest in sex, and how these vary by gender, in the British population.

► The study included detailed assessment of a range of relationship context and attitudinal variables seldom included in previous population-based surveys.

► Information about lacking interest in sex was assessed with a single item, asking participants whether they had lacked interest in having sex for a period of ≥3 months in the past year. Those who reported this were also asked whether they experienced associated distress.

► The cross-sectional data do not allow us to establish the causal direction of associations between lacking interest in sex and variables of interest.

## INTRODUCTION

In Britain's third National Survey of Sexual Attitudes and Lifestyles (Natsal-3), lacking interest in sex was the most common sexual difficulty reported by both men and women.[1] Lacking interest in sex for ≥3 months in the past year was twice as common in women compared with men. When duration and symptom severity criteria are considered (ie, that symptoms last ≥6 months and occur 'very often' or 'always'), these prevalence estimates are much lower,[2] but the gender difference is maintained.

Researchers have paid more attention to problems of low sexual interest in women than in men.[3–5] Among men the predominant focus has been on erectile functioning and on physiological causes of lacking interest in sex such as hormonal status, rather than on psychosocial

BMJ

determinants. This lack of attention to male problems is reflected in recent revisions to the Diagnostic and Statistical Manual (DSM-5) classification of sexual disorders[6] which involved major changes to sexual arousal and desire disorder categories in women, but no substantive changes for male disorders.

Most but not all studies involving men have reported an association between low sexual interest and increasing age (for review, see ref. 7). However, there are conflicting findings on the association with physical and mental health.[8 9] Limited research suggests that psychosocial and relationship factors may also be associated with low sexual desire in men.[8 10–12]

Among women, factors that have been consistently associated with lacking interest in sex are relationship problems, relationship quality and partner's sexual functioning,[13–17] poor physical health[18] and negative mood states/depression.[13 18 19] There are inconsistent findings on the association between low sexual interest and both age and menopausal status.[14 18] Few large-scale surveys have examined possible links between lacking interest in sex and either sexual attitudes or sexual behaviour. In the second wave of the British National Survey of Sexual Attitudes and Lifestyles (Natsal-2), among women, lacking interest in sex was associated with lower frequency of sex and attitudes according sex low priority.[20]

Studies have, for the most part, used small, clinical samples of patients seeking treatment for low sexual desire problems. The potential for bias in such studies is revealed in previously reported findings from Natsal-3 that only around a third of men and women with one or more sexual function problems meeting DSM-5 morbidity criteria had sought professional help in the last year.[2] The few large-scale probability-based surveys involving both men and women have focused on associations between low sexual desire and sociodemographic factors.

In summary, the evidence on the factors associated with men's and women's reports of low sexual desire is drawn largely from non-representative samples, is somewhat equivocal and, in men, sparse. Given that most previous research has involved non-representative samples, it is important to explore how correlates might differ in a population-based sample. Understanding the correlates of lacking interest in sex is key to informing therapeutic options for this group.

The research questions addressed in this paper are[1]: What sociodemographic, relationship, sexual behaviour and sexual attitudinal factors are associated with lacking interest in sex in sexually active men and women?[2] To what extent do these factors vary by gender?[3] To what extent does lacking interest in sex coexist with other sexual function problems?

## METHOD
### Participants and procedure
Natsal-3 is a probability sample survey of 15 162 men and women aged 16–74 years in Britain, interviewed between September 2010 and August 2012. A multistage,

clustered and stratified probability sample design was used and participants were interviewed in their homes by professional interviewers using a combination of computer-assisted personal interviews and computer-assisted self-interviews (CASIs) for the more sensitive questions (including, of relevance to this paper, those on sexual function). Interviewers were present in the room while participants completed the CASI, but did not view responses.[20] After weighting to adjust for unequal probabilities of selection and to match the British population in terms of age, gender and geographical region, the Natsal-3 sample was broadly representative, on key variables, of the British population as described by the 2011 Census.[21]

The estimated response rate was 57.7%, and the estimated cooperation rate (the number of interviews completed from eligible addresses for which contact was made) was 65.8% (of all eligible addressed contacted).[22] More extensive details of the survey methodology and sample characteristics are published elsewhere[21 22] and for demographic characteristics of the sample, see ref. 22. Participants provided oral informed consent for interviews and the survey was approved by the NRES Committee South-Central— Oxford A (ref.: 10/H0604/27).

Only respondents who reported ≥1 sexual partner (opposite-sex or same-sex) in the past year (4839 men and 6669 women) were asked whether they had lacked interest in sex for a period of ≥3 months in the past year (see below). These participants were the focus of the current analyses.

### Outcome measures
Items were drawn from the Natsal-SF, a measure of sexual function, designed and validated for population surveys. The measure comprises items on problems with sexual response, relational aspects of sexual function and self-appraisal of sex life.[23 24] Participants who reported at least one sexual partner in the past year (hereon 'sexually active participants') were asked, *In the last year, have you experienced any of the following for a period of ≥3 months?* and were given a list of difficulties and asked to indicate which they had experienced. The list included *Lacked interest in having sex.* Those indicating this difficulty were defined as lacking interest in having sex for a period of ≥3 months in the past year (the outcome for this analysis). Individuals reporting lacking interest in sex for at least 3 months were then asked, *And how do you feel about this?* with response options: *not at all distressed, a little distressed, fairly distressed* and *very distressed.* Those answering a little, fairly or very distressed were defined as lacking interest in sex and having distress about this symptom (outcome for sensitivity analysis, see below).

### Statistical analysis
All analyses were done using the complex survey functions of STATA V.14 to account for the weighting, clustering and stratification of the data. We used multivariable logistic regression to calculate age-adjusted ORs (AORs)

to examine the associations between reports of lacking interest in sex lasting ≥3 months in the past year, and sociodemographic, health, relationship, sexual behaviour and sexual attitude variables. For each variable, we also tested the interaction with gender to see if the magnitude of the associations between the above factors and reports of lacking interest in sex was the same for men and women. We conducted a sensitivity analysis for the outcome variable reporting lack of interest in sex lasting ≥3 months *and* distress about this symptom to assess whether similar associations were found. We also examined the association between reporting lacking interest in sex and the other sexual function problems asked about in Natsal-3 using AORs.

## RESULTS

Overall, 15.0% (95% CI 13.9% to 16.2%) of sexually active men and 34.2% (95% CI 32.8% to 35.5%) of sexually active women reported lacking interest in sex for ≥3 months in the year prior to interview. Table 1 presents the associations between lacking interest in sex and sociodemographic, health, relationship, sexual behaviour and sexual attitudinal variables for men and women.

Age was significantly associated with lacking interest in sex. Prevalence increased with age, being lowest among younger participants (16–24 years; men: 11.5%; women: 24.8%) and peaking in men aged 35–44 years (17.2%) and in women aged 55–64 years (38.8%). Regarding demographic variables, after adjusting for age, lack of interest was associated with leaving school at 16 (men only; AOR 1.31), being unemployed (men only; AOR: men: 1.44) and less frequent religious practice (women only; AOR 0.79). In women, after adjusting for age, those who were students or retired were less likely to lack desire.

After adjusting for age, there were associations between all physical and mental health variables assessed and lacking interest in sex. Individuals in poorer health (AORs: men: 3.29; women: 1.93), those who had much difficulty walking upstairs (AOR: men: 2.68; women: 1.55), those with a long-standing medical condition (AOR: men: 1.76; women: 1.35), and those who had screened positive for current depression (AOR: men: 2.95; women: 2.79) or who had been treated for depression in the past year (AOR: men: 2.82; women: 2.32) were more likely to report lacking interest in sex. The magnitude of these associations was similar for men and women. A greater number of comorbid health conditions was significantly associated with lacking interest in sex among both men and women. Menopausal status in women and circumcision in men were not associated with the likelihood of lacking sexual interest.

Regarding sexual behaviour, among both men and women, lack of interest was associated with frequency of sexual activity (defined as vaginal, oral or anal intercourse) in the four weeks prior to interview; 12.4% of men and 33.8% of women who reported having engaged in 3–4 sexual acts reported lack of interest versus 20.7%

of men and 42.9% of women who reported no sexual activity. Associations with recent masturbation differed by gender; lack of interest in sex was slightly *more* common among men who reported having recently masturbated but *less* common among women who did so. Women with three or more partners in the past year were less likely to report low sexual interest than those with only one partner (AOR 0.70) but there was no association between partner numbers and lacking interest in sex in men. Among men only, those who reported ever having taken drugs to assist sexual performance were more likely to report lacking interest in sex (AOR 1.36). A similar magnitude association was seen for women (AOR 1.39); however, fewer women reported ever having taken drugs and the 95% CI therefore crosses 1.

Associations were found between lacking interest in sex and several relationship contextual variables and for many of these variables associations were stronger for women than for men. For both men and women, lack of interest was associated with relationship status; women living with a partner were more likely to lack interest in sex than those in other relationship categories (see table 1). For women, all relationship categories had lower AORs than living with partner. Duration of most recent sexual relationship was significantly associated with lacking interest in sex only among women, being more common among those in longer relationships.

Among both men and women, there was an association between ease of communication and lacking interest in sex. Those who found it 'always easy to talk about sex' with their partner were less likely to report low interest. Lack of interest was more likely among those whose partner had sexual difficulties in the last year, and those who reported a lower assessment of happiness with the relationship, and not feeling emotionally close to partner during sex. Among women but not men, not sharing the same level of sexual interest with a partner, and not sharing the same sexual likes and dislikes, was also associated.

Having been pregnant in the last year was associated with lacking sexual interest as was having one or more young child(ren) (women only). Lack of interest in sex was significantly associated with sexual health indicators, including previous sexually transmitted infection (STI) diagnosis and ever having experienced non-volitional sex. The strength and direction of associations was similar for men and women, except for reporting another sexual function problem, which was significant for two or more problems in men, but one or more problems in women. Sexual competence at first sex was significantly associated with lack of interest in sex only among women.

Regarding attitudinal variables, both men and women who endorsed statements that 'people are under pressure to have sex' and 'people want less sex as they age' were more likely to report lacking interest in sex over the past year. The only attitudinal variable that showed a significant interaction with gender was that which related to men having a 'naturally higher sex drive than women'. Men who agreed with this statement were *less* likely than

**Table 1** Factors associated with lacking interest in having sex for at least 3 months in the past year in sexually active men and women

| | Men | | | | | | Women | | | | | | p Value for interaction with sex* |
| --- | --- | --- | --- | --- | --- | --- | --- | --- | --- | --- | --- | --- | --- |
| | Denom. (unwt, wt) | % | (95% CI) | Age-adjusted OR | (95% CI) | p Value | Denom. (unwt, wt) | % | (95% CI) | Age-adjusted OR | (95% CI) | p Value | |
| All | 4839, 5973 | 15.0 | (13.9 to 16.2) | | | | 6669, 5755 | 34.2 | (32.8 to 35.5) | | | | 0.6733 |
| *Sociodemographics* | | | | | | | | | | | | | |
| Age group (years) | | | | | | 0.0471 | | | | | | <0.0001 | |
| 16–24 | 1279, 936 | 11.5 | (9.4 to 14.0) | 1 | – | | 1662, 923 | 24.8 | (22.5 to 27.1) | 1 | – | | |
| 25–34 | 1376, 1238 | 14.6 | (12.7 to 16.6) | 1.32 | (1.00 to 1.73) | | 2236, 1246 | 31.9 | (29.8 to 34.1) | 1.42 | (1.22 to 1.66) | | |
| 35–44 | 719, 1298 | 17.2 | (14.5 to 20.4) | 1.61 | (1.19 to 2.18) | | 1050, 1290 | 36.8 | (33.7 to 40.1) | 1.77 | (1.48 to 2.13) | | |
| 45–54 | 630, 1186 | 15.3 | (12.5 to 18.7) | 1.40 | (1.01 to 1.95) | | 871, 1186 | 37.9 | (34.5 to 41.5) | 1.86 | (1.53 to 2.25) | | |
| 55–64 | 512, 849 | 16.5 | (13.4 to 20.2) | 1.53 | (1.10 to 2.13) | | 569, 755 | 38.8 | (34.5 to 43.2) | 1.92 | (1.55 to 2.39) | | |
| 65–74 | 323, 467 | 13.9 | (10.4 to 18.3) | 1.22 | (0.81 to 1.82) | | 281, 355 | 34.2 | (28.4 to 40.5) | 1.58 | (1.18 to 2.12) | | |
| Index of Multiple Deprivation (quintiles)† | | | | | | 0.093 | | | | | | 0.0316 | 0.0111 |
| 1 (least deprived) | 977, 1279 | 13.9 | (11.6 to 16.6) | 1 | – | | 1248, 1208 | 35.7 | (32.6 to 38.9) | 1 | – | | |
| 2 | 962, 1264 | 13.0 | (10.8 to 15.6) | 0.93 | (0.69 to 1.25) | | 1290, 1208 | 33.6 | (30.6 to 36.7) | 0.92 | (0.76 to 1.13) | | |
| 3 | 942, 1169 | 18.0 | (15.2 to 21.2) | 1.38 | (1.04 to 1.85) | | 1299, 1116 | 30.1 | (27.2 to 33.2) | 0.81 | (0.66 to 0.99) | | |
| 4 | 967, 1184 | 15.3 | (12.8 to 18.3) | 1.15 | (0.86 to 1.55) | | 1384, 1137 | 35.9 | (33.0 to 39.0) | 1.08 | (0.89 to 1.30) | | |
| 5 (most deprived) | 991, 1077 | 15.1 | (12.7 to 17.8) | 1.14 | (0.85 to 1.52) | | 1448, 1086 | 35.3 | (32.4 to 38.3) | 1.06 | (0.87 to 1.28) | | |
| Education level‡ | | | | | | 0.0083 | | | | | | 0.2453 | 0.2914 |
| Left school aged 17+ | 2862, 3464 | 13.5 | (12.1 to 15.1) | 1 | – | | 4150, 3406 | 32.7 | (31.0 to 34.5) | 1 | – | | |
| Left school at 16 | 1873, 2437 | 17.2 | (15.3 to 19.4) | 1.31 | (1.07 to 1.60) | | 2409, 2287 | 36.6 | (34.4 to 38.9) | 1.08 | (0.95 to 1.23) | | |
| Employment status | | | | | | 0.0086 | | | | | | 0.0003 | 0.0766 |
| Employed | 3211, 4254 | 14.7 | (13.3 to 16.1) | 1 | – | | 3871, 3517 | 34.6 | (32.9 to 36.4) | 1 | – | | |

Continued

**Table 1** Continued

| | Men | | | | | | Women | | | | | | p Value for interaction with sex* |
|---|---|---|---|---|---|---|---|---|---|---|---|---|---|
| | Denom. (unwt, wt) | % | (95% CI) | Age-adjusted OR | (95% CI) | p Value | Denom. (unwt, wt) | % | (95% CI) | Age-adjusted OR | (95% CI) | p Value | |
| Full-time education | 542, 431 | 12.6 | (8.8 to 17.5) | 0.98 | (0.64 to 1.51) | | 693, 423 | 22.5 | (19.0 to 26.4) | 0.70 | (0.55 to 0.89) | | |
| Unemployed | 707, 723 | 19.6 | (16.3 to 23.4) | 1.44 | (1.12 to 1.86) | | 1681, 1282 | 36.1 | (33.4 to 39.0) | 1.11 | (0.96 to 1.28) | | |
| Retired | 375, 562 | 13.6 | (10.4 to 17.7) | 0.75 | (0.52 to 1.09) | | 415, 524 | 35.8 | (31.0 to 40.9) | 0.75 | (0.57 to 0.99) | | |
| Practises religion at least once a month | | | | | | 0.1687 | | | | | | 0.0082 | 0.9966 |
| No | 4283, 5179 | 15.3 | (14.1 to 16.6) | 1 | – | | 5659, 4754 | 34.8 | (33.3 to 36.3) | 1 | – | | |
| Yes | 521, 748 | 12.9 | (10.0 to 16.4) | 0.81 | (0.60 to 1.09) | | 956, 945 | 30.7 | (27.5 to 34.2) | 0.79 | (0.67 to 0.94) | | |
| *Health* | | | | | | | | | | | | | |
| Self-reported general health | | | | | | <0.0001 | | | | | | <0.0001 | 0.1890 |
| Very good/good | 4123, 5055 | 13.4 | (12.2 to 14.6) | 1 | – | | 5683, 4851 | 32.3 | (30.9 to 33.8) | 1 | – | | |
| Fair | 580, 745 | 21.9 | (18.3 to 25.8) | 1.8 | (1.41 to 2.30) | | 780, 709 | 42.2 | (38.2 to 46.3) | 1.45 | (1.21 to 1.75) | | |
| Bad/very bad | 135, 171 | 33.9 | (25.3 to 43.6) | 3.29 | (2.14 to 5.06) | | 206, 195 | 49.9 | (42.2 to 57.7) | 1.93 | (1.40 to 2.67) | | |
| Difficulty walking up stairs because of a health problem | | | | | | <0.0001 | | | | | | 0.0497 | 0.1179 |
| No difficulty | 4475, 5460 | 14.1 | (12.9 to 15.3) | 1 | – | | 6062, 5107 | 33.3 | (31.8 to 34.7) | 1 | – | | |
| Some difficulty | 278, 393 | 23.0 | (18.1 to 28.8) | 1.8 | (1.30 to 2.49) | | 450, 482 | 39.2 | (34.4 to 44.2) | 1.15 | (0.92 to 1.43) | | |
| Much difficulty/ unable to do this | 86, 120 | 30.9 | (20.9 to 43.0) | 2.68 | (1.57 to 4.57) | | 157, 166 | 47.0 | (38.0 to 56.1) | 1.55 | (1.06 to 2.25) | | |
| Long-standing illness or disability | | | | | | <0.0001 | | | | | | <0.0001 | 0.1348 |
| No | 3585, 4259 | 12.8 | (11.6 to 14.2) | 1 | – | | 4843, 4026 | 31.6 | (30.0 to 33.2) | 1 | – | | |
| Yes | 1253, 1713 | 20.5 | (18.1 to 23.1) | 1.76 | (1.44 to 2.16) | | 1825, 1729 | 40.1 | (37.5 to 42.8) | 1.35 | (1.17 to 1.55) | | |

**Table 1** Continued

| | Men | | | | | | Women | | | | | | p Value for interaction with sex* |
|---|---|---|---|---|---|---|---|---|---|---|---|---|---|
| | Denom. (unwt, wt) | % | (95% CI) | Age-adjusted OR | (95% CI) | p Value | Denom. (unwt, wt) | % | (95% CI) | Age-adjusted OR | (95% CI) | p Value | |
| Number of comorbid conditions§ | | | | | | <0.0001 | | | | | | <0.0001 | 0.7951 |
| 0 | 3453, 3994 | 12.8 | (11.5 to 14.1) | 1 | – | | 4357, 3536 | 29.9 | (28.2 to 31.5) | 1 | – | | |
| 1 | 939, 1329 | 18.9 | (16.2 to 21.9) | 1.64 | (1.30 to 2.06) | | 1555, 1416 | 38.6 | (35.9 to 41.5) | 1.42 | (1.23 to 1.64) | | |
| ≥2 | 446, 650 | 21.0 | (17.0 to 25.6) | 1.91 | (1.41 to 2.60) | | 755, 802 | 45.1 | (41.2 to 49.1) | 1.75 | (1.45 to 2.13) | | |
| Depressive symptoms¶ | | | | | | <0.0001 | | | | | | <0.0001 | 0.6249 |
| No | 4383, 5471 | 13.5 | (12.4 to 14.8) | 1 | – | | 5885, 5149 | 31.7 | (30.2 to 33.1) | 1 | – | | |
| Yes | 449, 495 | 31.3 | (26.4 to 36.7) | 2.95 | (2.26 to 3.85) | | 780, 602 | 55.2 | (51.0 to 59.5) | 2.79 | (2.32 to 3.37) | | |
| Treated for depression, past year | | | | | | <0.0001 | | | | | | <0.0001 | 0.2447 |
| No | 4524, 5630 | 14.0 | (12.9 to 15.2) | 1 | – | | 5770, 5040 | 31.7 | (30.2 to 33.2) | 1 | – | | |
| Yes | 313, 342 | 31.5 | (25.7 to 38.0) | 2.82 | (2.08 to 3.83) | | 897, 713 | 51.4 | (47.6 to 55.2) | 2.32 | (1.96 to 2.75) | | |
| Menopausal status | | | | | | | | | | | | 0.9326 | |
| Not menopausal | | | | | | | 5485, 4187 | 32.3 | (30.9 to 33.8) | 1 | – | | |
| Menopausal | | | | | | | 1167, 1548 | 38.9 | (36.0 to 41.9) | 0.99 | (0.79 to 1.24) | | |
| Circumcised | | | | | | 0.5951 | | | | | | | |
| No | 3909, 4728 | 15.1 | (13.8 to 16.4) | 1 | – | | | | | | | | |
| Yes | 857, 1166 | 14.5 | (12.0 to 17.4) | 0.94 | (0.73 to 1.20) | | | | | | | | |
| Sexual behaviour | | | | | | <0.0001 | | | | | | <0.0001 | 0.4778 |
| Number of occasions of sex, past four weeks | | | | | | | | | | | | | |
| 0 | 1013, 1163 | 20.7 | (17.8 to 23.8) | 1 | – | | 1408, 1245 | 42.9 | (39.9 to 45.9) | 1 | – | | |

Continued

**Table 1** Continued

| | Men | | | | | | Women | | | | | | p Value for interaction with sex* |
|---|---|---|---|---|---|---|---|---|---|---|---|---|---|
| | Denom. (unwt, wt) | % | (95% CI) | Age-adjusted OR | (95% CI) | p Value | Denom. (unwt, wt) | % | (95% CI) | Age-adjusted OR | (95% CI) | p Value | |
| 1–2 | 1160, 1566 | 18.7 | (16.2 to 21.5) | 0.89 | (0.69 to 1.14) | | 1481, 1373 | 39.6 | (36.7 to 42.5) | 0.89 | (0.75 to 1.05) | | |
| 3–4 | 870, 1168 | 12.4 | (10.1 to 15.1) | 0.54 | (0.41 to 0.73) | | 1240, 1130 | 33.8 | (30.7 to 37.0) | 0.7 | (0.58 to 0.85) | | |
| 5+ | 1617, 1869 | 9.2 | (7.8 to 11.0) | 0.39 | (0.30 to 0.51) | | 2078, 1655 | 22.6 | (20.5 to 24.8) | 0.41 | (0.34 to 0.49) | | |
| Masturbation, past four weeks | | | | | | 0.0458 | | | | | | 0.0038 | 0.0005 |
| No | 1297, 1828 | 13.7 | (11.8 to 15.8) | 1 | – | | 4032, 3612 | 36.0 | (34.3 to 37.7) | 1 | – | | |
| Yes | 3531, 4132 | 15.6 | (14.2 to 17.0) | 1.24 | (1.00 to 1.52) | | 2615, 2114 | 30.8 | (28.7 to 33.0) | 0.83 | (0.73 to 0.94) | | |
| Number of sexual partners, past year** | | | | | | 0.5348 | | | | | | 0.0038 | 0.0183 |
| 1 | 3573, 4824 | 15.0 | (13.7 to 16.3) | 1 | – | | 5440, 5012 | 35.3 | (33.8 to 36.8) | 1 | – | | |
| 2 | 539, 513 | 16.2 | (12.9 to 20.3) | 1.14 | (0.86 to 1.52) | | 570, 364 | 28.2 | (23.9 to 32.8) | 0.80 | (0.63 to 1.01) | | |
| 3+ | 718, 627 | 13.6 | (11.1 to 16.6) | 0.94 | (0.72 to 1.22) | | 642, 366 | 24.8 | (21.0 to 29.0) | 0.70 | (0.56 to 0.88) | | |
| Paid for sex, past year | | | | | | 0.7167 | | | | | | | |
| No | 4774, 5896 | 15.0 | (13.9 to 16.2) | 1 | – | | | | | | | | |
| Yes | 64, 75 | 13.4 | (6.8 to 24.7) | 0.87 | (0.41 to 1.84) | | | | | | | | |
| Ever taken drugs to assist sexual performance | | | | | | 0.0175 | | | | | | 0.0666 | 0.8967 |
| No | 4188, 5180 | 14.4 | (13.2 to 15.7) | 1 | – | | 6478, 5624 | 34.0 | (32.6 to 35.4) | 1 | – | | |
| Yes | 636, 776 | 19.0 | (15.7 to 22.8) | 1.36 | (1.06 to 1.76) | | 184, 124 | 40.0 | (32.0 to 48.5) | 1.39 | (0.98 to 1.96) | | |
| *Relationship context* | | | | | | | | | | | | | |
| Relationship status | | | | | | 0.0383 | | | | | | <0.0001 | 0.0001 |
| Living with partner | 2708, 4266 | 15.5 | (14.1 to 17.1) | 1 | – | | 3967, 4168 | 37.9 | (36.3 to 39.7) | 1 | – | | |

Continued

**Table 1** Continued

| | Men | | | | | | Women | | | | | | p Value for interaction with sex* |
|---|---|---|---|---|---|---|---|---|---|---|---|---|---|
| | Denom. (unwt, wt) | % | (95% CI) | Age-adjusted OR | (95% CI) | p Value | Denom. (unwt, wt) | % | (95% CI) | Age-adjusted OR | (95% CI) | p Value | |
| In a steady relationship, not living together | 947, 760 | 12.0 | (9.6 to 14.8) | 0.76 | (0.57 to 1.00) | | 1360, 790 | 22.6 | (20.2 to 25.2) | 0.51 | (0.43 to 0.60) | | |
| Not in a steady relationship, but previously cohabited | 446, 388 | 18.2 | (14.6 to 22.5) | 1.22 | (0.91 to 1.62) | | 752, 462 | 28.9 | (25.4 to 32.8) | 0.68 | (0.56 to 0.83) | | |
| Not in a steady relationship, never cohabited | 727, 551 | 12.4 | (9.9 to 15.5) | 0.8 | (0.58 to 1.09) | | 580, 330 | 21.3 | (17.6 to 25.5) | 0.49 | (0.38 to 0.63) | | |
| Duration of most recent sexual relationship (years) | | | | | | 0.494 | | | | | | <0.0001 | <0.0001 |
| ≤1 | 1462, 1260 | 13.0 | (11.0 to 15.3) | 1 | – | | 1597, 998 | 21.5 | (19.1 to 24.1) | 1 | – | | |
| Between 1 and 5 | 1247, 1227 | 15.3 | (13.2 to 17.7) | 1.21 | (0.94 to 1.55) | | 1758, 1148 | 28.5 | (26.1 to 31.0) | 1.45 | (1.20 to 1.76) | | |
| Between 5 and 15 | 1065, 1484 | 14.9 | (12.6 to 17.5) | 1.14 | (0.86 to 1.50) | | 1774, 1458 | 39.8 | (37.2 to 42.4) | 2.37 | (1.96 to 2.86) | | |
| >15 | 1004, 1904 | 16.1 | (13.9 to 18.7) | 1.19 | (0.87 to 1.63) | | 1445, 2036 | 40.0 | (37.3 to 42.7) | 2.31 | (1.84 to 2.91) | | |
| Always easy to talk about sex with partners†† | | | | | | 0.0001 | | | | | | <0.0001 | 0.0182 |
| Yes | 1695, 1899 | 11.5 | (9.7 to 13.5) | 1 | – | | 1746, 1451 | 22.6 | (20.4 to 25.1) | 1 | – | | |
| No/other | 3122, 4048 | 16.7 | (15.3 to 18.2) | 1.53 | (1.23 to 1.90) | | 4907, 4289 | 38.0 | (36.4 to 39.6) | 2.06 | (1.77 to 2.39) | | |
| Happy with relationship‡‡ | | | | | | <0.0001 | | | | | | <0.0001 | 0.8679 |
| Yes | 1951, 2791 | 12.6 | (11.0 to 14.4) | 1 | – | | 2736, 2601 | 31.5 | (29.5 to 33.6) | 1 | – | | |
| Other | 995, 1430 | 21.0 | (18.4 to 23.9) | 1.85 | (1.47 to 2.32) | | 1640, 1617 | 45.4 | (42.7 to 48.1) | 1.79 | (1.55 to 2.08) | | |
| Participant does not share same level of interest in sex as partner | | | | | | 0.2339 | | | | | | <0.0001 | <0.0001 |

Continued

**Table 1** Continued

| | Men | | | | | | Women | | | | | | p Value for interaction with sex* |
|---|---|---|---|---|---|---|---|---|---|---|---|---|---|
| | Denom. (unwt, wt) | % | (95% CI) | Age-adjusted OR | (95% CI) | p Value | Denom. (unwt, wt) | % | (95% CI) | Age-adjusted OR | (95% CI) | p Value | |
| No/other | 2270, 3233 | 15.0 | (13.4 to 16.7) | 1 | – | | 3211, 3064 | 27.2 | (25.4 to 29.0) | 1 | – | | |
| Yes | 676, 988 | 17.1 | (14.2 to 20.4) | 1.17 | (0.90 to 1.51) | | 1166, 1155 | 62.5 | (59.2 to 65.7) | 4.57 | (3.87 to 5.38) | | |
| Participant does not share same sexual likes and dislikes as partner | | | | | | 0.4188 | | | | | | <0.0001 | <0.0001 |
| No/other | 2650, 3803 | 15.3 | (13.8 to 16.9) | 1 | – | | 4079, 3908 | 34.9 | (33.3 to 36.6) | 1 | – | | |
| Yes | 296, 418 | 17.3 | (13.0 to 22.5) | 1.16 | (0.81 to 1.66) | | 297, 310 | 61.0 | (54.6 to 67.2) | 2.91 | (2.22 to 3.83) | | |
| Partner experienced sexual difficulties in the past year | | | | | | 0.0136 | | | | | | <0.0001 | 0.4140 |
| No/other | 2431, 3454 | 14.6 | (13.1 to 16.2) | 1 | – | | 3726, 3498 | 34.8 | (33.1 to 36.6) | 1 | – | | |
| Yes | 513, 763 | 19.4 | (15.8 to 23.6) | 1.41 | (1.07 to 1.86) | | 649, 719 | 46.8 | (42.5 to 51.1) | 1.60 | (1.32 to 1.94) | | |
| Does not feel emotionally close to partner when having sex | | | | | | 0.0006 | | | | | | <0.0001 | 0.5972 |
| No/other | 2904, 4165 | 15.1 | (13.7 to 16.6) | 1 | – | | 4263, 4108 | 35.9 | (34.3 to 37.6) | 1 | – | | |
| Yes | 42, 56 | 39.9 | (23.6 to 58.8) | 3.74 | (1.76 to 7.93) | | 112, 109 | 73.0 | (62.8 to 81.3) | 4.80 | (2.99 to 7.69) | | |
| *Lifestyle* | | | | | | | | | | | | | |
| 1+ child(ren) aged <5 in household | | | | | | 0.9088 | | | | | | <0.0001 | 0.0216 |
| No, none | 4100, 5015 | 15.2 | (13.9 to 16.5) | 1 | – | | 4997, 4671 | 33.1 | (31.6 to 34.6) | 1 | – | | |
| Yes, 1+ | 727, 941 | 14.5 | (11.9 to 17.6) | 0.98 | (0.76 to 1.28) | | 1664, 1074 | 38.6 | (36.0 to 41.4) | 1.55 | (1.34 to 1.79) | | |
| Pregnant in the last year | | | | | | | | | | | | 0.0114 | |
| No | | | | | | | 4227, 4122 | 36.2 | (34.6 to 37.9) | 1 | – | | |

Continued

**Table 1** Continued

| | Men | | | | | | Women | | | | | | p Value for interaction with sex* |
|---|---|---|---|---|---|---|---|---|---|---|---|---|---|
| | Denom. (unwt, wt) | % | (95% CI) | Age-adjusted OR | (95% CI) | p Value | Denom. (unwt, wt) | % | (95% CI) | Age-adjusted OR | (95% CI) | p Value | |
| Yes | | | | | | | 437, 273 | 41.7 | (36.6 to 47.1) | 1.36 | (1.07 to 1.72) | | |
| Used hormonal contraceptive, past year | | | | | | | | | | | | 0.05 | |
| No | | | | | | | 3759, 3838 | 34.8 | (33.1 to 36.5) | 1 | – | | |
| Yes | | | | | | | 2806, 1831 | 33.0 | (30.9 to 35.1) | 1.15 | (1.00 to 1.33) | | |
| *Sexual health indicators* | | | | | | | | | | | | | |
| Ever diagnosed with a sexually transmitted infection | | | | | | <0.0001 | | | | | | 0.0004 | 0.0651 |
| No (or only thrush) | 4147, 5127 | 14.0 | (12.8 to 15.3) | 1 | – | | 5455, 4861 | 33.4 | (31.9 to 34.9) | 1 | – | | |
| Yes (excluding thrush) | 677, 830 | 21.4 | (18.1 to 25.0) | 1.67 | (1.33 to 2.10) | | 1206, 888 | 38.2 | (35.1 to 41.5) | 1.32 | (1.13 to 1.54) | | |
| Ever experienced non-volitional sex | | | | | | 0.0010 | | | | | | <0.0001 | 0.3164 |
| No | 4705, 5824 | 14.7 | (13.6 to 16.0) | 1 | – | | 5815, 5055 | 32.8 | (31.4 to 34.2) | 1 | – | | |
| Yes/don't know | 133, 148 | 26.1 | (18.9 to 34.9) | 2.07 | (1.34 to 3.18) | | 848, 695 | 44.3 | (40.5 to 48.3) | 1.66 | (1.40 to 1.97) | | |
| Sexual competence at first sex§§ | | | | | | 0.0706 | | | | | | <0.0001 | 0.1797 |
| Not competent | 2407, 3037 | 16.2 | (14.6 to 17.9) | 1 | – | | 3438, 2927 | 37.6 | (35.7 to 39.5) | 1 | – | | |
| Competent | 2302, 2784 | 13.7 | (12.1 to 15.4) | 0.84 | (0.69 to 1.01) | | 3097, 2716 | 30.3 | (28.4 to 32.3) | 0.73 | (0.65 to 0.83) | | |
| Number of other sexual response problems experienced¶¶ | | | | | | <0.0001 | | | | | | <0.0001 | 0.0015 |
| 0 | 3208, 3945 | 11.7 | (10.5 to 13.1) | 1 | – | | 4377, 3759 | 25.3 | (23.8 to 26.9) | 1 | – | | |
| 1 | 1061, 1350 | 10.9 | (9.0 to 13.2) | 0.91 | (0.71 to 1.17) | | 1217, 1087 | 34.8 | (31.7 to 38.0) | 1.55 | (1.32 to 1.82) | | |

**Table 1** Continued

| | Men | | | | | | Women | | | | | | p Value for interaction with sex* |
|---|---|---|---|---|---|---|---|---|---|---|---|---|---|
| | Denom. (unwt, wt) | % | (95% CI) | Age-adjusted OR | (95% CI) | p Value | Denom. (unwt, wt) | % | (95% CI) | Age-adjusted OR | (95% CI) | p Value | |
| 2+ | 570, 678 | 42.5 | (37.9 to 47.2) | 5.58 | (4.41 to 7.04) | | 1075, 909 | 69.8 | (66.5 to 72.9) | 6.91 | (5.82 to 8.21) | | |
| *Attitudes* | | | | | | | | | | | | | |
| People are under pressure to have sex | | | | | | 0.0115 | | | | | | 0.0001 | 0.7970 |
| Else | 1799, 2264 | 13.1 | (11.4 to 15.0) | 1 | – | | 1851, 1570 | 29.3 | (26.8 to 31.9) | 1 | – | | |
| Strongly agree/agree | 3038, 3707 | 16.2 | (14.7 to 17.8) | 1.29 | (1.06 to 1.57) | | 4817, 4185 | 36.0 | (34.4 to 37.6) | 1.34 | (1.16 to 1.54) | | |
| People want less sex as they age | | | | | | <0.0001 | | | | | | <0.0001 | 0.9443 |
| Else | 2943, 3472 | 11.4 | (10.2 to 12.8) | 1 | – | | 4044, 3278 | 27.8 | (26.2 to 29.4) | 1 | – | | |
| Strongly agree/agree | 1894, 2499 | 20.0 | (18.0 to 22.2) | 1.93 | (1.61 to 2.32) | | 2624, 2477 | 42.6 | (40.4 to 44.8) | 1.85 | (1.63 to 2.10) | | |
| Men have a naturally higher sex drive than women | | | | | | <0.0001 | | | | | | <0.0001 | <0.0001 |
| Else | 2788, 3441 | 18.0 | (16.4 to 19.7) | 1 | – | | 3351, 2830 | 26.0 | (24.3 to 27.8) | 1 | – | | |
| Strongly agree/agree | 2049, 2530 | 10.9 | (9.4 to 12.6) | 0.56 | (0.46 to 0.68) | | 3317, 2925 | 42.0 | (40.0 to 44.1) | 2.04 | (1.80 to 2.31) | | |
| Too much sex in the media | | | | | | 0.7069 | | | | | | 0.1807 | 0.4835 |
| Else | 1986, 2296 | 14.6 | (12.8 to 16.6) | 1 | – | | 2091, 1618 | 31.7 | (29.3 to 34.2) | 1 | – | | |
| Strongly agree/agree | 2851, 3675 | 15.3 | (13.8 to 16.9) | 1.04 | (0.85 to 1.26) | | 4577, 4137 | 35.1 | (33.5 to 36.8) | 1.10 | (0.96 to 1.26) | | |

Continued

## Table 1 Continued

| | Men | | | | Women | | | | p Value for interaction with sex* |
|---|---|---|---|---|---|---|---|---|---|
| | Denom. (unwt, wt) | % | (95% CI) | Age-adjusted OR | (95% CI) | p Value | Denom. (unwt, wt) | % | (95% CI) | Age-adjusted OR | (95% CI) | p Value | |

Denominator is those aged 16–74 years with at least one partner in the past year.

*p Value for interaction to determine whether the magnitude of association between each variable and lack of interest in sex differs between men and women.

†Index of Multiple Deprivation (IMD) is a multidimensional measure of area (neighbourhood)-level deprivation based on the participant's postcode. IMD scores for England, Scotland and Wales were adjusted before being combined and assigned to quintiles, using a method by Payne and Abel.[50]

‡Participants aged≥17 years.

§Includes arthritis, heart attack, coronary heart disease, angina, other forms of heart disease, hypertension, stroke, diabetes, broken hip or pelvis, bone or hip replacement ever, backache lasting >3 months, any other muscle or bone disease lasting >3 months, depression, cancer and any thyroid condition treated in the past year.

¶Participants were asked whether they had often been bothered by feeling down, depressed or hopeless in the past two weeks and whether they had often been bothered by little interest or pleasure in doing things in the past two weeks, using a validated two-question patient health questionnaire (PHQ-2).

**Opposite and/or same-sex partners.

††Other means easy with a husband or wife or regular partner, but difficult with a new partner; easy with a new partner, but difficult with a husband or wife or regular partner; difficult with any partner; it depends, sometimes easy and sometimes difficult.

‡‡Participants were asked to rate how happy they were in their relationship from 1 (very happy) to 7 (very unhappy); responses of 1 or 2 were regarded as denoting participants who were happy with their relationship.

§§A constructed variable to measure readiness, combining consensuality, autonomy of decision-making, timing and use of effective contraception.

¶¶Sexual response problems (for at least 3 months in the past year): lacked enjoyment in sex, felt anxious during sex, felt physical pain as a result of sex, felt no excitement or arousal during sex, difficulty in reaching climax, reached climax more quickly than you would like, trouble getting or keep an erection (men), uncomfortably dry vagina (women).

Unwt, unweighted; wt, weighted.

---

those who disagreed to lack interest in sex, while the reverse was true among women.

Table 2 presents the associations between lacking interest in sex and being distressed about this (as a measure/marker of severity), and the above sociodemographic, health and sexual relationship/behaviour variables. While prevalence was lower, the associations and the interactions with gender were generally similar; however, some of the previous gender-specific associations with variables (eg, masturbation, and pregnancy in women, and education in men) were no longer significant when the outcome variable was reported low sexual interest *and* associated distress. In addition, some associations became stronger when considering only those who reported a distressing lack of interest in sex (vs lack of interest without any reported distress). For example, the association between depressive symptoms and having been treated for depression in the past year was stronger in men than in women.

Regarding the association between reporting lacking interest in sex and the other sexual function problems asked about in Natsal-3, the strongest (positive) associations were for lacking enjoyment in sex (AOR 9.78 and 8.95 for men and women, respectively), followed by feeling no excitement or arousal during sex (AOR 9.21 and 9.16 for men and women, respectively) (see table 3).

## DISCUSSION

We identified a broad range of factors, including some that have not been explored in previous large-scale surveys, that were associated with men's and women's reports of lacking interest in sex in a representative British population-based survey. Our findings, discussed below, revealed some gender similarities as well as some interesting gender differences. The strongest evidence for gender differences was for the relationship context variables, where associations with lacking interest in sex were much stronger for women than for men.

### Interpretation of findings in context of previous research

Our finding relating to differences by age is consistent with some, but not all, results from previous research which has yielded generally inconsistent findings. Some studies have, like ours, shown a higher prevalence of sexual interest problems in older than in younger women.[25–27] Others have found no association between age and low sexual interest complaints[14 28] and yet more have shown lack of sexual interest to be more common among younger women.[18] Whereas we found a marginal relationship with age in men, some studies (though not all, eg, ref. 29) have found a stronger relationship.[12 30] It is possible that the varied findings might in part be a result of varied definitions of low sexual interest or differences in sampling.

The finding in this analysis that having young children appears to increase the likelihood of reporting lack of sexual interest for women, but not for men, remains

**Table 2** Factors associated with lacking interest in having sex for at least 3 months in the past year and being distressed about it in sexually active men and women

| | Men | | | | | | Women | | | | | | p Value for interaction with sex* |
|---|---|---|---|---|---|---|---|---|---|---|---|---|---|
| | Denom. (unwt, wt) | % | (95% CI) | Age-adjusted OR | (95% CI) | p Value | Denom. (unwt, wt) | % | (95% CI) | Age-adjusted OR | (95% CI) | p Value | |
| All | 4839, 5973 | 8.2 | (7.4 to 9.1) | | | | 6669, 5755 | 20.8 | (19.6 to 22.0) | | | | |
| *Sociodemographics* | | | | | | | | | | | | | |
| Age group (years) | | | | | | 0.0011 | | | | | | <0.0001 | 0.8971 |
| 16–24 | 1279, 936 | 4.8 | (3.7 to 6.4) | 1 | – | | 1662, 923 | 15.2 | (13.4 to 17.3) | 1 | – | | |
| 25–34 | 1376, 1238 | 8.0 | (6.7 to 9.5) | 1.7 | (1.19 to 2.41) | | 2236, 1246 | 20.9 | (19.0 to 22.8) | 1.47 | (1.22 to 1.76) | | |
| 35–44 | 719, 1298 | 9.6 | (7.5 to 12.3) | 2.09 | (1.40 to 3.13) | | 1050, 1290 | 22.9 | (20.3 to 25.7) | 1.65 | (1.34 to 2.04) | | |
| 45–54 | 630, 1186 | 9.7 | (7.4 to 12.6) | 2.11 | (1.38 to 3.22) | | 871, 1186 | 23.3 | (20.4 to 26.6) | 1.69 | (1.35 to 2.13) | | |
| 55–64 | 512, 849 | 9.4 | (7.0 to 12.6) | 2.04 | (1.30 to 3.21) | | 569, 755 | 21.8 | (18.3 to 25.8) | 1.55 | (1.20 to 2.01) | | |
| 65–74 | 323, 467 | 5.5 | (3.4 to 8.6) | 1.13 | (0.65 to 1.99) | | 281, 355 | 16.5 | (12.4 to 21.7) | 1.10 | (0.76 to 1.59) | | |
| Index of Multiple Deprivation (quintiles)† | | | | | | 0.8339 | | | | | | 0.0938 | 0.4592 |
| 1 (least deprived) | 977, 1279 | 8.1 | (6.2 to 10.4) | 1 | – | | 1248, 1208 | 23.3 | (20.7 to 26.1) | 1 | – | | |
| 2 | 962, 1264 | 7.4 | (5.7 to 9.6) | 0.92 | (0.62 to 1.36) | | 1290, 1208 | 20.8 | (18.2 to 23.5) | 0.87 | (0.69 to 1.09) | | |
| 3 | 942, 1169 | 8.3 | (6.4 to 10.6) | 1.05 | (0.71 to 1.55) | | 1299, 1116 | 19.6 | (17.1 to 22.4) | 0.82 | (0.65 to 1.03) | | |
| 4 | 967, 1184 | 8.8 | (6.9 to 11.1) | 1.14 | (0.78 to 1.66) | | 1384, 1137 | 21.9 | (19.3 to 24.7) | 0.95 | (0.76 to 1.18) | | |
| 5 (most deprived) | 991, 1077 | 8.6 | (6.7 to 10.9) | 1.12 | (0.75 to 1.65) | | 1448, 1086 | 18.2 | (15.9 to 20.6) | 0.75 | (0.60 to 0.94) | | |
| Education level‡ | | | | | | 0.4958 | | | | | | 0.7324 | 0.4496 |
| Left school aged 17+ | 2862, 3464 | 7.9 | (6.8 to 9.2) | 1 | – | | 4150, 3406 | 20.8 | (19.4 to 22.4) | 1 | – | | |
| Left school at 16 | 1873, 2437 | 8.8 | (7.5 to 10.4) | 1.09 | (0.85 to 1.41) | | 2409, 2287 | 21.1 | (19.3 to 23.1) | 0.97 | (0.83 to 1.14) | | |
| Employment status | | | | | | 0.0001 | | | | | | 0.0003 | 0.1244 |
| Employed | 3211, 4254 | 8.3 | (7.3 to 9.5) | 1 | – | | 3871, 3517 | 21.6 | (20.1 to 23.2) | 1 | – | | |
| Full-time education | 542, 431 | 4.9 | (2.8 to 8.6) | 0.74 | (0.38 to 1.44) | | 693, 423 | 14.8 | (11.8 to 18.4) | 0.75 | (0.56 to 1.01) | | |
| Unemployed | 707, 723 | 12.1 | (9.5 to 15.3) | 1.56 | (1.14 to 2.13) | | 1681, 1282 | 22.3 | (19.9 to 24.9) | 1.07 | (0.90 to 1.27) | | |
| Retired | 375, 562 | 4.9 | (3.1 to 7.6) | 0.41 | (0.23 to 0.71) | | 415, 524 | 16.8 | (13.4 to 20.8) | 0.57 | (0.41 to 0.79) | | |
| Practises religion at least once a month | | | | | | 0.1638 | | | | | | 0.0167 | 0.8143 |
| No | 4283, 5179 | 8.5 | (7.5 to 9.5) | 1 | – | | 5659, 4754 | 21.5 | (20.2 to 22.9) | 1 | – | | |
| Yes | 521, 748 | 6.4 | (4.4 to 9.4) | 0.73 | (0.48 to 1.13) | | 956, 945 | 18.0 | (15.3 to 20.9) | 0.78 | (0.63 to 0.96) | | |
| *Health* | | | | | | | | | | | | | |
| Self-reported general health | | | | | | <0.0001 | | | | | | <0.0001 | 0.0969 |
| Very good/good | 4123, 5055 | 7.0 | (6.1 to 7.9) | 1 | – | | 5683, 4851 | 19.2 | (18.0 to 20.5) | 1 | – | | |
| Fair | 580, 745 | 13.3 | (10.5 to 16.8) | 2.04 | (1.50 to 2.78) | | 780, 709 | 27.9 | (24.3 to 31.9) | 1.60 | (1.30 to 1.97) | | |
| Bad/very bad | 135, 171 | 22.6 | (15.3 to 32.1) | 3.85 | (2.31 to 6.40) | | 206, 195 | 33.4 | (26.3 to 41.4) | 2.05 | (1.45 to 2.91) | | |
| Difficulty walking up stairs because of a health problem | | | | | | 0.0001 | | | | | | 0.0085 | 0.1553 |
| No difficulty | 4475, 5460 | 7.6 | (6.7 to 8.6) | 1 | – | | 6062, 5107 | 20.1 | (18.9 to 21.4) | 1 | – | | |
| Some difficulty | 278, 393 | 12.3 | (8.8 to 17.0) | 1.67 | (1.11 to 2.52) | | 450, 482 | 24.1 | (20.0 to 28.7) | 1.21 | (0.93 to 1.59) | | |

Continued

**Table 2** Continued

| | Men | | | | | | Women | | | | | | p Value for interaction with sex* |
|---|---|---|---|---|---|---|---|---|---|---|---|---|---|
| | Denom. (unwt, wt) | % | (95% CI) | Age-adjusted OR | (95% CI) | p Value | Denom. (unwt, wt) | % | (95% CI) | Age-adjusted OR | (95% CI) | p Value | |
| Much difficulty/unable to do this | 86, 120 | 22.2 | (13.5 to 34.2) | 3.36 | (1.79 to 6.32) | | 157, 166 | 32.3 | (24.3 to 41.5) | 1.81 | (1.21 to 2.70) | | 0.0345 |
| Long-standing illness or disability | | | | | | <0.0001 | | | | | | <0.0001 | |
| No | 3585, 4259 | 6.5 | (5.6 to 7.5) | 1 | – | | 4843, 4026 | 18.7 | (17.4 to 20.0) | 1 | – | | |
| Yes | 1253, 1713 | 12.5 | (10.6 to 14.8) | 2.09 | (1.60 to 2.74) | | 1825, 1729 | 25.7 | (23.4 to 28.2) | 1.48 | (1.27 to 1.74) | | |
| Number of comorbid conditions§ | | | | | | <0.0001 | | | | | | <0.0001 | 0.5779 |
| 0 | 3453, 3994 | 6.4 | (5.5 to 7.5) | 1 | – | | 4357, 3536 | 17.3 | (15.9 to 18.7) | 1 | – | | |
| 1 | 939, 1329 | 11.0 | (9.0 to 13.4) | 1.88 | (1.37 to 2.57) | | 1555, 1416 | 24.1 | (21.7 to 26.7) | 1.54 | (1.30 to 1.83) | | |
| ≥2 | 446, 650 | 13.3 | (10.1 to 17.4) | 2.40 | (1.61 to 3.59) | | 755, 802 | 30.5 | (26.8 to 34.4) | 2.16 | (1.74 to 2.69) | | |
| Depressive symptoms¶ | | | | | | <0.0001 | | | | | | <0.0001 | 0.0370 |
| No | 4383, 5471 | 6.8 | (6.0 to 7.7) | 1 | – | | 5885, 5149 | 18.6 | (17.4 to 19.8) | 1 | – | | |
| Yes | 449, 495 | 23.7 | (19.3 to 28.9) | 4.36 | (3.20 to 5.94) | | 780, 602 | 39.6 | (35.4 to 44.0) | 2.94 | (2.41 to 3.59) | | |
| Treated for depression, past year | | | | | | <0.0001 | | | | | | <0.0001 | 0.0371 |
| No | 4524, 5630 | 7.3 | (6.5 to 8.2) | 1 | – | | 5770, 5040 | 18.5 | (17.3 to 19.8) | 1 | – | | |
| Yes | 313, 342 | 23.0 | (17.9 to 29.1) | 3.81 | (2.71 to 5.36) | | 897, 713 | 36.4 | (32.9 to 40.2) | 2.54 | (2.12 to 3.03) | | |
| Menopausal status | | | | | | | | | | | | 0.9656 | |
| Not menopausal | | | | | | | 5485, 4187 | 20.2 | (18.9 to 21.5) | 1 | – | | |
| Menopausal | | | | | | | 1167, 1548 | 22.5 | (20.0 to 25.2) | 1.01 | (0.76 to 1.32) | | |
| Circumcised | | | | | | 0.4097 | | | | | | | |
| No | 3909, 4728 | 8.3 | (7.4 to 9.4) | 1 | – | | | | | | | | |
| Yes | 857, 1166 | 7.5 | (5.7 to 9.9) | 0.87 | (0.62 to 1.22) | | | | | | | | |
| Sexual behaviour | | | | | | | | | | | | | |
| Number of occasions of sex, past four weeks | | | | | | <0.0001 | | | | | | <0.0001 | 0.5496 |
| 0 | 1013, 1163 | 10.3 | (8.3 to 12.7) | 1 | – | | 1408, 1245 | 23.2 | (20.7 to 26.0) | 1 | – | | |
| 1–2 | 1160, 1566 | 10.5 | (8.6 to 12.8) | 1.02 | (0.74 to 1.42) | | 1481, 1373 | 24.2 | (21.8 to 26.9) | 1.06 | (0.87 to 1.30) | | |
| 3–4 | 870, 1168 | 7.4 | (5.6 to 9.8) | 0.71 | (0.48 to 1.04) | | 1240, 1130 | 21.3 | (18.7 to 24.2) | 0.91 | (0.73 to 1.13) | | |
| 5+ | 1617, 1869 | 5.0 | (3.9 to 6.4) | 0.46 | (0.33 to 0.66) | | 2078, 1655 | 14.7 | (12.9 to 16.7) | 0.58 | (0.47 to 0.72) | | |
| Masturbation, past four weeks | | | | | | 0.0164 | | | | | | 0.7265 | 0.0309 |
| No | 1297, 1828 | 6.9 | (5.5 to 8.6) | 1 | – | | 4032, 3612 | 21.1 | (19.6 to 22.6) | 1 | – | | |
| Yes | 3531, 4132 | 8.8 | (7.7 to 9.9) | 1.42 | (1.07 to 1.88) | | 2615, 2114 | 20.3 | (18.4 to 22.2) | 0.97 | (0.84 to 1.13) | | |
| Number of sexual partners, past year** | | | | | | 0.2466 | | | | | | 0.0016 | 0.4744 |
| 1 | 3573, 4824 | 8.5 | (7.5 to 9.6) | 1 | – | | 5440, 5012 | 21.6 | (20.3 to 22.9) | 1 | – | | |
| 2 | 539, 513 | 6.3 | (4.3 to 9.1) | 0.75 | (0.49 to 1.14) | | 570, 364 | 16.7 | (13.3 to 20.6) | 0.75 | (0.57 to 0.99) | | |

Continued

**Table 2** Continued

| | Men | | | | | | Women | | | | | | p Value for interaction with sex* |
|---|---|---|---|---|---|---|---|---|---|---|---|---|---|
| | Denom. (unwt, wt) | % | (95% CI) | Age-adjusted OR | (95% CI) | p Value | Denom. (unwt, wt) | % | (95% CI) | Age-adjusted OR | (95% CI) | p Value | |
| 3+ | 718, 627 | 6.8 | (5.1 to 9.0) | 0.82 | (0.59 to 1.15) | | 642, 366 | 14.1 | (11.0 to 17.7) | 0.62 | (0.46 to 0.83) | | |
| Paid for sex, past year | | | | | | 0.4865 | | | | | | | |
| No | 4774, 5896 | 8.2 | (7.4 to 9.2) | 1 | – | | | | | | | | |
| Yes | 64, 75 | 5.6 | (1.8 to 16.4) | 0.66 | (0.20 to 2.15) | | | | | | | | |
| Ever taken drugs to assist sexual performance | | | | | | 0.0022 | | | | | | 0.1055 | 0.5305 |
| No | 4188, 5180 | 7.6 | (6.7 to 8.6) | 1 | – | | 6478, 5624 | 20.6 | (19.5 to 21.8) | 1 | – | | |
| Yes | 636, 776 | 12.1 | (9.5 to 15.4) | 1.63 | (1.19 to 2.23) | | 184, 124 | 25.9 | (19.2 to 33.9) | 1.38 | (0.93 to 2.05) | | |
| Relationship status | | | | | | 0.03 | | | | | | <0.0001 | 0.0307 |
| Living with partner | 2708, 4266 | 8.8 | (7.7 to 10.1) | 1 | – | | 3967, 4168 | 23.4 | (21.9 to 24.9) | 1 | – | | |
| In a steady relationship, not living together | 947, 760 | 6.9 | (5.3 to 9.0) | 0.78 | (0.56 to 1.09) | | 1360, 790 | 15.4 | (13.4 to 17.7) | 0.59 | (0.49 to 0.71) | | |
| Not in a steady relationship, but previously cohabited | 446, 388 | 8.8 | (6.2 to 12.2) | 1.00 | (0.67 to 1.48) | | 752, 462 | 13.6 | (11.1 to 16.6) | 0.51 | (0.40 to 0.66) | | |
| Not in a steady relationship, never cohabited | 727, 551 | 4.7 | (3.3 to 6.8) | 0.52 | (0.34 to 0.81) | | 580, 330 | 11.0 | (8.2 to 14.5) | 0.39 | (0.28 to 0.55) | | |
| Duration of most recent sexual relationship (years) | | | | | | 0.0143 | | | | | | <0.0001 | 0.0719 |
| ≤1 | 1462, 1260 | 5.5 | (4.3 to 7.1) | 1 | – | | 1597, 998 | 11.2 | (9.4 to 13.2) | 1 | – | | |
| Between 1 and 5 | 1247, 1227 | 9.0 | (7.3 to 11.0) | 1.67 | (1.18 to 2.36) | | 1758, 1148 | 18.5 | (16.5 to 20.7) | 1.81 | (1.44 to 2.29) | | |
| Between 5 and 15 | 1065, 1484 | 9.3 | (7.5 to 11.6) | 1.68 | (1.17 to 2.43) | | 1774, 1458 | 25.2 | (23.0 to 27.6) | 2.81 | (2.23 to 3.55) | | |
| >15 | 1004, 1904 | 8.8 | (7.1 to 10.8) | 1.47 | (0.97 to 2.22) | | 1445, 2036 | 23.8 | (21.5 to 26.2) | 2.83 | (2.13 to 3.75) | | |
| Always easy to talk about sex with partners†† | | | | | | 0 | | | | | | <0.0001 | 0.4854 |
| Yes | 1695, 1899 | 4.8 | (3.8 to 6.0) | 1 | – | | 1746, 1451 | 11.4 | (9.7 to 13.2) | 1 | – | | |
| No/other | 3122, 4048 | 9.8 | (8.7 to 11.1) | 2.15 | (1.62 to 2.87) | | 4907, 4289 | 23.9 | (22.5 to 25.3) | 2.43 | (2.02 to 2.93) | | |
| Happy with relationship‡‡ | | | | | | <0.0001 | | | | | | <0.0001 | 0.9717 |
| Yes | 1951, 2791 | 7.1 | (5.9 to 8.6) | 1 | – | | 2736, 2601 | 18.6 | (16.9 to 20.4) | 1 | – | | |
| Other | 995, 1430 | 13.3 | (11.2 to 15.8) | 2.01 | (1.51 to 2.66) | | 1640, 1617 | 31.4 | (28.8 to 34.0) | 2.00 | (1.69 to 2.37) | | |
| Participant does not share same level of interest in sex as partner | | | | | | 0.0975 | | | | | | 0.0311 | <0.0001 |
| No/other | 2270, 3233 | 8.5 | (7.2 to 10.0) | 1 | – | | 3211, 3064 | 15.0 | (13.6 to 16.4) | 1 | – | | |
| Yes | 676, 988 | 11.6 | (9.2 to 14.4) | 1.41 | (1.03 to 1.92) | | 1166, 1155 | 46.2 | (42.9 to 49.6) | 4.91 | (4.13 to 5.83) | | |
| Participant does not share same sexual likes and dislikes as partner | | | | | | 0.0027 | | | | | | <0.0001 | 0.0212 |
| No/other | 2650, 3803 | 8.9 | (7.7 to 10.2) | 1 | – | | 4079, 3908 | 22.1 | (20.6 to 23.6) | 1 | – | | |
| Yes | 296, 418 | 12.2 | (8.6 to 17.0) | 1.43 | (0.94 to 2.18) | | 297, 310 | 41.9 | (35.6 to 48.6) | 2.55 | (1.93 to 3.37) | | |
| Partner experienced sexual difficulties in the past year | | | | | | | | | | | | <0.0001 | 0.6889 |

Continued

**Table 2** Continued

| | Men | | | | | | Women | | | | | | p Value for interaction with sex* |
|---|---|---|---|---|---|---|---|---|---|---|---|---|---|
| | Denom. (unwt, wt) | % | (95% CI) | Age-adjusted OR | (95% CI) | p Value | Denom. (unwt, wt) | % | (95% CI) | Age-adjusted OR | (95% CI) | p Value | |
| No/other | 2431, 3454 | 8.3 | (7.2 to 9.6) | 1 | – | | 3726, 3498 | 22.1 | (20.6 to 23.7) | 1 | – | | |
| Yes | 513, 763 | 13.2 | (10.2 to 17.0) | 1.68 | (1.20 to 2.35) | | 649, 719 | 30.4 | (26.5 to 34.6) | 1.58 | (1.27 to 1.95) | | 0.8228 |
| Does not feel emotionally close to partner when having sex | | | | | | 0.0225 | | | | | | <0.0001 | |
| No/other | 2904, 4165 | 9.1 | (7.9 to 10.3) | 1 | – | | 4263, 4108 | 22.9 | (21.5 to 24.4) | 1 | – | | |
| Yes | 42, 56 | 21.0 | (10.2 to 38.3) | 2.69 | (1.15 to 6.29) | | 112, 109 | 47.0 | (36.4 to 57.8) | 2.98 | (1.92 to 4.63) | | 0.0042 |
| *Lifestyle* | | | | | | | | | | | | | |
| 1+child(ren) aged<5 in household | | | | | | 0.1047 | | | | | | 0.0004 | |
| No, none | 4100, 5015 | 8.6 | (7.6 to 9.6) | 1 | – | | 4997, 4671 | 20.2 | (18.9 to 21.5) | 1 | – | | |
| Yes, 1+ | 727, 941 | 6.3 | (4.6 to 8.5) | 0.75 | (0.52 to 1.06) | | 1664, 1074 | 23.5 | (21.2 to 25.9) | 1.34 | (1.14 to 1.58) | | |
| Pregnant in the last year | | | | | | | | | | | | 0.5927 | |
| No | | | | | | | 4227, 4122 | 21.8 | (20.4 to 23.4) | 1 | – | | |
| Yes | | | | | | | 437, 273 | 20.7 | (16.6 to 25.6) | 0.92 | (0.69 to 1.24) | | |
| Used hormonal contraceptive, past year | | | | | | | | | | | | 0.1141 | |
| No | | | | | | | 3759, 3838 | 20.7 | (19.2 to 22.3) | 1 | – | | |
| Yes | | | | | | | 2806, 1831 | 20.9 | (19.1 to 22.7) | 1.14 | (0.97 to 1.35) | | |
| *Sexual health indicators* | | | | | | | | | | | | | |
| Ever diagnosed with a sexually transmitted infection | | | | | | <0.0001 | | | | | | 0.0002 | |
| No (or only thrush) | 4148, 5128 | 7.3 | (6.5 to 8.3) | 1 | – | | 5455, 4861 | 20.0 | (18.7 to 21.3) | 1 | – | | |
| Yes (excluding thrush) | 677, 830 | 13.7 | (11.0 to 17.0) | 2.02 | (1.51 to 2.70) | | 1206, 888 | 25.1 | (22.3 to 28.1) | 1.39 | (1.16 to 1.65) | | 0.0291 |
| Ever experienced non-volitional sex | | | | | | <0.0001 | | | | | | <0.0001 | |
| No | 4706, 5825 | 7.9 | (7.1 to 8.9) | 1 | – | | 5815, 5055 | 19.4 | (18.2 to 20.7) | 1 | – | | |
| Yes/don't know | 133, 148 | 19.4 | (13.1 to 27.7) | 2.83 | (1.74 to 4.59) | | 848, 695 | 30.9 | (27.3 to 34.6) | 1.86 | (1.55 to 2.25) | | 0.1143 |
| Sexual competence at first sex§§ | | | | | | 0.4876 | | | | | | <0.0001 | |
| Not competent | 2408, 3039 | 8.7 | (7.5 to 10.0) | 1 | – | | 3438, 2927 | 23.6 | (21.9 to 25.3) | 1 | – | | |
| Competent | 2302, 2784 | 7.8 | (6.6 to 9.2) | 0.91 | (0.71 to 1.18) | | 3097, 2716 | 17.7 | (16.1 to 19.3) | 0.70 | (0.61 to 0.81) | | 0.0787 |
| Number of other sexual response problems experienced¶¶ | | | | | | <0.0001 | | | | | | <0.0001 | |
| 0 | 3209, 3947 | 5.3 | (4.4 to 6.3) | 1 | – | | 4377, 3759 | 12.9 | (11.7 to 14.1) | 1 | – | | |
| 1 | 1061, 1350 | 6.1 | (4.7 to 7.8) | 1.14 | (0.81 to 1.59) | | 1217, 1087 | 21.7 | (19.0 to 24.6) | 1.86 | (1.53 to 2.26) | | |
| 2+ | 570, 678 | 29.7 | (25.4 to 34.4) | 7.57 | (5.68 to 10.10) | | 1075, 909 | 52.4 | (48.9 to 56.0) | 7.48 | (6.25 to 8.94) | | 0.0262 |
| *Attitudes* | | | | | | | | | | | | | |
| People are under pressure to have sex | | | | | | 0.1437 | | | | | | <0.0001 | 0.2192 |

Continued

**Table 2** Continued

| | Men | | | | | | Women | | | | | | p Value for interaction with sex* |
|---|---|---|---|---|---|---|---|---|---|---|---|---|---|
| | Denom. (unwt, wt) | % | (95% CI) | Age-adjusted OR | (95% CI) | p Value | Denom. (unwt, wt) | % | (95% CI) | Age-adjusted OR | (95% CI) | p Value | |
| Else | 1799, 2264 | 7.4 | (6.0 to 9.0) | 1 | – | | 1851, 1570 | 16.4 | (14.5 to 18.5) | 1 | – | | |
| Strongly agree/agree | 3038, 3707 | 8.7 | (7.6 to 9.9) | 1.21 | (0.94 to 1.57) | | 4817, 4185 | 22.4 | (21.0 to 23.9) | 1.47 | (1.24 to 1.74) | | |
| People want less sex as they age | | | | | | 0.0005 | | | | | | <0.0001 | 0.8045 |
| Else | 2943, 3472 | 6.7 | (5.7 to 7.8) | 1 | – | | 4044, 3278 | 17.1 | (15.8 to 18.6) | 1 | – | | |
| Strongly agree/agree | 1894, 2499 | 10.3 | (8.8 to 12.1) | 1.58 | (1.22 to 2.04) | | 2624, 2477 | 25.6 | (23.7 to 27.6) | 1.64 | (1.43 to 1.90) | | |
| Men have a naturally higher sex drive than women | | | | | | <0.0001 | | | | | | <0.0001 | <0.0001 |
| Else | 2788, 3441 | 10.2 | (8.9 to 11.5) | 1 | – | | 3351, 2830 | 15.9 | (14.4 to 17.4) | 1 | – | | |
| Strongly agree/agree | 2049, 2530 | 5.5 | (4.4 to 6.9) | 0.52 | (0.39 to 0.68) | | 3317, 2925 | 25.5 | (23.8 to 27.4) | 1.81 | (1.56 to 2.09) | | |
| Too much sex in the media | | | | | | 0.3477 | | | | | | 0.0693 | 0.8856 |
| Else | 1986, 2296 | 7.5 | (6.3 to 9.0) | 1 | – | | 2091, 1618 | 18.8 | (16.8 to 20.9) | 1 | – | | |
| Strongly agree/agree | 2851, 3675 | 8.6 | (7.5 to 9.9) | 1.13 | (0.88 to 1.46) | | 4577, 4137 | 21.6 | (20.2 to 23.0) | 1.16 | (0.99 to 1.36) | | |

Denominator is those aged 16–74 years with at least one partner in the past year.

*p Value for interaction to determine whether the magnitude of association between each variable and lack of interest in sex differs between men and women.

†Index of Multiple Deprivation (IMD) is a multidimensional measure of area (neighbourhood)-level deprivation based on the participant's postcode. IMD scores for England, Scotland and Wales were adjusted before being combined and assigned to quintiles, using a method by Payne and Abel.[50]

‡Participants aged≥17 years.

§Includes arthritis, heart attack, coronary heart disease, angina, other forms of heart disease, hypertension, stroke, diabetes, broken hip or pelvis, bone or hip replacement ever, backache lasting >3months, any other muscle or bone disease lasting >3months, depression, cancer and any thyroid condition treated in the past year.

¶Participants were asked whether they had often been bothered by feeling down, depressed or hopeless in the past twoweeks and whether they had often been bothered by little interest or pleasure in doing things in the past twoweeks, using a validated two-question patient health questionnaire (PHQ-2).

**Opposite and/or same-sex partners.

††Other means easy with a husband or wife or regular partner, but difficult with a new partner; easy with a new partner; difficult with any partner; difficult with a husband or wife or regular partner; it depends, sometimes easy and sometimes difficult.

‡‡Participants were asked to rate how happy they were in their relationship from 1 (very happy) to 7 (very unhappy); responses of 1 or 2 were regarded as denoting participants who were happy with their relationship.

§§A constructed variable to measure readiness, combining consensuality, autonomy of decision-making, timing and use of effective contraception.

¶¶Sexual response problems (for at least 3 months in the past year): lacked enjoyment in sex, felt anxious during sex, felt no excitement or arousal during sex, difficulty in reaching climax, reached a climax more quickly than you would like, trouble getting or keep an erection (men), uncomfortably dry vagina (women).

Unwt, unweighted; wt, weighted.

**Table 3** Associations between reporting lack of interest in having sex for at least 3 months in the past year and other sexual response problems lasting ≥3 months in the past year, by sex

| Denominators (unwt, wt) | Men | | | | | Women | | | | |
|---|---|---|---|---|---|---|---|---|---|---|
| | Did not report a lack of interest in sex | Reported a lack of interest in sex | AOR* | (95% CI) | p Value | Did not report a lack of interest in sex | Reported a lack of interest in sex | AOR* | (95% CI) | p Value |
| | 4126, 5077 | 713, 897 | | | | 4540, 3790 | 2129, 1965 | | | |
| **Lacked enjoyment in having sex** | | | | | <0.0001 | | | | | <0.0001 |
| No | 97.7% (97.1 to 98.1) | 81.5% (78.2 to 84.4) | 1 | – | | 95.9% (95.1 to 96.5) | 72.5% (70.2 to 74.7) | 1 | – | |
| Yes | 2.3% (1.9 to 2.9) | 18.5% (15.6 to 21.8) | 9.78 | (7.11 to 13.46) | | 4.1% (3.5 to 4.9) | 27.5% (25.3 to 29.8) | 8.95 | (7.28 to 11.01) | |
| **Felt anxious during sex** | | | | | <0.0001 | | | | | <0.0001 |
| No | 96.1% (95.5 to 96.7) | 85.8% (82.6 to 88.5) | 1 | – | | 97.3% (96.7 to 97.7) | 89.9% (88.4 to 91.3) | 1 | – | |
| Yes | 3.9% (3.3 to 4.5) | 14.2% (11.5 to 17.4) | 4.16 | (3.08 to 5.62) | | 2.7% (2.3 to 3.3) | 10.1% (8.7 to 11.6) | 4.4 | (3.43 to 5.65) | |
| **Felt physical pain as a result of sex** | | | | | 0.0213 | | | | | <0.0001 |
| No | 98.4% (97.9 to 98.8) | 97.1% (95.6 to 98.1) | 1 | – | | 95.7% (95.0 to 96.3) | 86.5% (84.6 to 88.1) | 1 | – | |
| Yes | 1.6% (1.2 to 2.1) | 2.9% (1.9 to 4.4) | 1.87 | (1.10 to 3.19) | | 4.3% (3.7 to 5.0) | 13.5% (11.9 to 15.4) | 3.55 | (2.83 to 4.45) | |
| **Felt no excitement or arousal during sex** | | | | | <0.0001 | | | | | <0.0001 |
| No | 98.5% (98.0 to 98.9) | 87.7% (85.0 to 90.0) | 1 | – | | 97.5% (96.9 to 97.9) | 80.9% (79.0 to 82.7) | 1 | – | |
| Yes | 1.5% (1.1 to 2.0) | 12.3% (10.0 to 15.0) | 9.21 | (6.33 to 13.40) | | 2.5% (2.1 to 3.1) | 19.1% (17.3 to 21.0) | 9.16 | (7.16 to 11.70) | |
| **Difficulty in reaching climax** | | | | | <0.0001 | | | | | <0.0001 |
| No | 92.7% (91.7 to 93.5) | 80.5% (76.6 to 83.8) | 1 | – | | 88.3% (87.2 to 89.3) | 74.9% (72.7 to 76.9) | 1 | – | |
| Yes | 7.3% (6.5 to 8.3) | 19.5% (16.2 to 23.4) | 3.08 | (2.37 to 3.99) | | 11.7% (10.7 to 12.8) | 25.1% (23.1 to 27.3) | 2.6 | (2.23 to 3.03) | |

Continued

**Table 3** Continued

| Denominators (unwt, wt) | Men | | | | | Women | | | | |
|---|---|---|---|---|---|---|---|---|---|---|
| | Did not report a lack of interest in sex | Reported a lack of interest in sex | AOR* | (95% CI) | p Value | Did not report a lack of interest in sex | Reported a lack of interest in sex | AOR* | (95% CI) | p Value |
| | 4126, 5077 | 713, 897 | | | | 4540, 3790 | 2129, 1965 | | | |
| Reached climax more quickly than you would like | | | | | 0.0198 | | | | | 0.3658 |
| No | 85.6% (84.3 to 86.9) | 82.0% (78.7 to 85.0) | 1 | – | | 97.8% (97.2 to 98.2) | 97.5% (96.7 to 98.1) | 1 | – | |
| Yes | 14.4% (13.1 to 15.7) | 18.0% (15.0 to 21.3) | 1.32 | (1.05 to 1.68) | | 2.2% (1.8 to 2.8) | 2.5% (1.9 to 3.3) | 1.18 | (0.82 to 1.69) | |
| Trouble getting or keeping an erection | | | | | <0.0001 | | | | | |
| No | 88.5% (87.3 to 89.6) | 79.4% (75.9 to 82.6) | 1 | – | | | | | | |
| Yes | 11.5% (10.4 to 12.7) | 20.6% (17.4 to 24.1) | 1.97 | (1.55 to 2.51) | | | | | | |
| Uncomfortably dry vagina | | | | | | | | | | <0.0001 |
| No | | | | | | 90.7% (89.5 to 91.7) | 80.1% (77.9 to 82.1) | 1 | – | |
| Yes | | | | | | 9.3% (8.3 to 10.5) | 19.9% (17.9 to 22.1) | 2.28 | (1.89 to 2.76) | |

Denominator is those aged 16–74 years with at least one partner in the past year.
*AOR comparing those reporting lacking interest to those who did not.
Unwt, unweighted; wt, weighted.

unchanged since the previous Natsal-2 survey.[31] This may be due to fatigue associated with a primary caring role,[32] the fact that daily stress appears to affect sexual functioning in women more than men[33] or possibly a shift in focus of attention attendant on bringing up small children.

The finding of a link between lacking interest in sex and lacking enjoyment in sex and/or feeling no excitement or arousal during sex is not surprising and has been shown in previous studies.[3] The strong associations between lack of interest in sex and physical and mental health indicators, which we observed for both men and women, are not entirely consistent with findings from other studies. While this link has been persuasively shown for women,[13 18 19] in men, the evidence is more equivocal. In a study of men attending an outpatient clinic for sexual problems, psychological symptoms such as anxiety and depression were more predictive of low sexual desire than hormonal or other physical markers.[11] In contrast, DeRogatis et al,[9] in their study of men with erectile dysfunction, observed no differences in depressive symptoms, concurrent illness or medication use between men with and without symptoms of low sexual desire.

The gender differences in associations between masturbation and a lack of sexual interest are interesting and have been explored in few previous population-based studies. Our observation that lack of interest was *more* commonly reported by men who had recently masturbated, but *less* commonly reported by women who had done so, may reflect a tendency among women for self-pleasuring to be, not a substitute for partnered sex but instead a part of a broader repertoire of sexual fulfilment; this possibility is worthy of further exploration. In contrast, for men frequency of masturbation reflects reduced frequency of partnered sex.[34] However, it is worth noting that in the U.S. National Health and Social Life Survey lifetime number of sexual partners and masturbation practices were unrelated to the likelihood of sexual desire difficulties for either men or women.[35]

Our observation that duration of most recent sexual relationship showed a strong association with lacking interest in sex in women is consistent with previous studies.[15 17] There has been little comparable research on men with which to corroborate the absence of such an association among men in our analysis.

Our data confirm the importance of the relational context in individuals' level of sexual interest. The strong associations between relationship and partner factors and sexual interest are consistent with those shown in many previous studies relating to women[13–17] and with a much smaller literature in men.[36 37] In particular, sexual dysfunction in a male partner has previously been associated with women's levels of sexual desire,[15 38 39] and sexual desire discrepancy in couples has been linked to lower reported relationship satisfaction and more couple conflict.[40]

The strong links found between several key sexual health outcomes and lack of interest in sex are interesting; among both men and women, reporting an STI diagnosis and non-volitional sex were associated with reporting lack of interest in sex. Our finding that lacking 'sexual competence' at first sexual intercourse was linked with subsequent lack of interest in sex among women but not men may reflect a greater salience of contextual aspects of first sex for women. More women than men report being pressured by a partner on the first occasion of heterosexual intercourse, and to have subsequently experienced regret about first sexual experiences.[41] These findings suggest that for women early sexual experiences may shape future sexual encounters/relationships to a greater extent than for men.

To our knowledge, no previous studies have assessed the association between attitudes towards sexual matters and lack of interest in sex. Endorsing the assumption that 'people want less sex as they age' was associated with lack of interest in both genders. It might be that this belief contributes to a decline in interest, or—equally plausible—that those who lack interest adopt this attitude to avoid viewing their experience as problematic. Interestingly, men who endorsed the view that 'men have a higher sex drive than women' were significantly *less* likely to report lacking interest in sex, whereas women who agreed with this statement were *more* likely to do so. If people responded to this statement with reference to their own relationship, these findings may be seen as making intuitive sense. The results suggest that endorsing stereotypical gender norms related to sex may adversely affect women more than men.

## Strengths and limitations

Strengths of our study include the use of national probability sample survey data involving both men and women across a wide age range.[21 22] With a few exceptions (eg, refs.12 14 29 42), most surveys on sexual desire problems have sampled either men *or* women, precluding direct comparisons within the same sample. Another strength was the detailed and holistic examination of relationship context and attitudinal variables, which few previous studies have reported. Response rates for Natsal-3 were also similar to those of other major social surveys in Britain[43] and higher than many previous surveys of sexual problems.[35 44]

Limitations include the cross-sectional nature of the data, which mean that we are unable to infer temporality and causality. The sample is representative of those resident in private households in Britain, that is, not those living in institutions. We included only respondents who reported ≥1 sexual partner (opposite-sex or same-sex) in the past year, excluding those who had not had sex because of lack of interest. We only used a single item to assess lacking interest in sex, although we additionally took account of whether those who reported this also reported that it caused them distress, as a way of trying to capture more problematic lack of interest. This sensitivity analysis enabled us to demonstrate that for most variables similar associations exist regardless of whether or not distress was reported. It is important to acknowledge,

however, that these data do not necessarily correspond to clinical diagnoses. Finally, we have tested many associations within this study and some will have been significant by chance. These were exploratory and descriptive analyses of zero-order relationships and therefore some of the smaller effect sizes may not replicate and may not hold in multivariable analyses.

### Implications for research and practice

The findings indicate that lack of interest in sex is associated with a broad range of factors across sociodemographic, relationship, sexual behaviour and sexual attitudinal domains. There are both research and clinical applications of our results.

First, our findings underscore the importance of the relational context in understanding low sexual interest in both men and women. For women in particular, the experience of sexual interest appears strongly linked with their perceptions of the quality of their relationships, their communication with partners and their expectations/attitudes about sex. Our findings support the view that transient (and often adaptive) reductions in sexual desire are not evidence of 'dysfunction'.[45]

In the context of the recent US Food and Drug Administration approval of flibanserin, the first drug to treat low sexual desire in women,[46] these findings are relevant to the current debate about whether striving for a pharmaceutical solution to women's sexual desire problems is an appropriate and feasible goal.[45 47] Some authors have suggested that women with complaints of low sexual interest might benefit most from integrative approaches that accord with a biopsychosocial model.[48]

Second, our findings on the strong association between open sexual communication (ie, 'finding it always easy to talk about sex') and a reduced likelihood of reporting lack of interest in sex, particularly for women, emphasise the importance of providing a broad sexual and relationships education, rather than limiting attention only to adverse consequences of sex and how to prevent them. Similarly, the important role of early sexual experiences, and sexual 'competence', especially for women, in shaping later experiences of sexual desire supports the need for comprehensive sex education.

In a clinical context, our findings emphasise the importance of healthcare professionals assessing psychological and interpersonal variables in individuals presenting with complaints of low sexual interest.[49] In couple therapy, it is important that therapists have an awareness of the differences between men and women in the factors associated with low sexual interest. For example, among the subgroup of participants reporting both lack of interest in sex and related distress, we found a stronger association between depressive symptoms and treatment for depression in the last year among men compared with women. Lastly, our findings support previous research on the critical role of physical and mental health in understanding low sexual interest problems experienced by men and women.[11 18]

## CONCLUSIONS

This study extends our understanding of the factors associated with lack of interest in sex in men and women, the gender similarities and differences, and highlights the need to assess and—if necessary—treat sexual desire problems in a holistic and relationship, as well as gender-specific way.

**Contributors** The paper was conceived by CAG, CHM, AMJ, KW and KRM. CAG wrote the first draft, with further contributions from all authors. Statistical analyses were undertaken by CHM, CT and KGJ. CHM, AMJ (principal investigator) and KW, initial applicants on Natsal-3, wrote the study protocol and obtained funding. Natsal-3 questionnaire design, ethics applications and piloting were undertaken by CHM, CT, AMJ, KW and KRM. Data management was undertaken by NatCen Social Research, UCL and LSHTM. All authors contributed to data interpretation, reviewed successive drafts and approved the final version of the manuscript.

**Funding** Natsal-3 was supported by grants from the U.K. Medical Research Council (G0701757) and the Wellcome Trust (084840), with support from the Economic and Social Research Council and the Department of Health. KM has been supported by the United Kingdom Medical Research Council grant MC_UU_12017/11 and Scottish Government Chief Scientist Office grant SPHSU11.

**Competing interests** AMJ has been a governor of the Wellcome Trust since 2011.

**Patient consent** Obtained.

**Ethics approval** Natsal-3 was approved by the NRES Committee South Central-Oxford A (Ref: 10/H0604/27).

**Provenance and peer review** Not commissioned; externally peer reviewed.

**Data sharing statement** The Natsal-3 data set is publicly available from the UK Data Service: https://discover.ukdataservice.ac.uk/; SN: 7799; persistent identifier: 10.5255/UKDA-SN-77991-1.

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
