## [Reviewer comments · BMJ Open]

ARTICLE DETAILS

TITLE (PROVISIONAL)	What factors are associated with reporting lacking interest in sex and how do these vary by gender?: Findings from the third British National Survey of Sexual Attitudes and Lifestyles
AUTHORS	Graham, Cynthia; Mercer, Catherine; Tanton, Clare; Jones, Kyle; Johnson, Anne; Wellings, Kaye; Mitchell, Kirstin

VERSION 1 - REVIEW

REVIEWER	Miriam K. Forbes, PhD University of Minnesota, USA
REVIEW RETURNED	28-Mar-2017

GENERAL COMMENTS	This paper uses the NATSAL-3 data to explicate the sociodemographic; mental and physical health; sexual health, behaviours and attitudes; relationship; and lifestyle factors associated with a lack of interest in sex that has lasted for three months or more in the past year. The NATSAL-3 data are ideal for these analyses, particularly with the use of methodological variables that account for the complex survey design. The aim of painting a detailed picture of the correlates of lacking sexual interest is also important (although I think the authors need to make this stronger case in-text). I would suggest that the manuscript could be strengthened by some changes in the statistical analyses and the reporting of results in particular, as described below. I have also listed some other questions, comments, and suggestions below that may help to strengthen the manuscript. Introduction 1. The intro is brief and to the point, but I think that it needs a paragraph to develop a rationale for why it is important to understand the correlates of lacking sexual interest. Method 2. These data are ideal for addressing the aims of the study. 3. Why were participants required to have a sexual partner in the past year to be included in this study? 4. The associations between lacking interest in sex and other sexual function problems seem tangential to the main aims of the study. Results and Discussion 5. Overall, I think the results need a little more attention to detail. For example, some of the significant results were not reported in-text: a. the index of multiple deprivation was a significant predictor for womenb. employment status was also significant for womenc. number of comorbid health conditions was a significant predictor for men and women
--

d. taking drugs to assist sexual performance was a predictor for men

e. relationship status was differentially related to lacking interest in sex for women (i.e., all categories had lower AORs than living with partner) vs men

f. non-volitional sex was a significant predictor for men and women

6. There are also some examples where multiple response categories were conflated in-text:

a. "at least three sexual acts" is 3-4 sexual acts specifically, with separate results (albeit the same pattern) for 5+

b. "women with two or more partners" (vs. 1) AOR is for 3+ partners

7. There are a few times where it needs to be specified that an effect was gender-specific:

a. pregnancy and children under 5

b. sexual competence at first sex

These three points (5-7) are obviously nitpicking --particularly given the number of results included in Table 1 and reported in-text-- but since the aim of the paper is to be descriptive, I think particularly careful attention to detail is warranted.

8. I also think it is important that the significance levels are adjusted for the multiple comparisons. It's good that the authors currently interpret the effect sizes (AORs), rather than purely the significance, but given the large number of predictors and the large sample size, a p-value of .05 will likely include a substantial number of false positive results. Similarly, the reported CIs should be proportionately widened.

9. It might also be interesting (although not necessary) to look at the unique predictions of the significant correlates - e.g., to answer questions like does relationship status predict lack of interest in sex over and above relationship duration?

10. There is some richness in the sensitivity analyses (Table 2) that may make the discussion more robust - for example, there are significant interactions for gender and the depression variables (stronger effects for men); women's masturbation is no longer associated with lack of interest in sex (but men's is); pregnancy is no longer a significant predictor, etc. In the context of considering lacking interest in sex in general vs. a distressing lack of interest in sex, these differences in the results are potentially interesting and worth spending time on.

11. If space is an issue (and it's an option) it might be worth combining the results and discussion to avoid the need to re-state the results in the discussion. Alternatively, referring readers to the table without spending time interpreting each result in-text (in the results section) might free up space to go into more detail for some points in the discussion.

12. The implications section opens with "Our findings underscore the importance of the relational context in understanding low sexual interest in both men and women" (p. 14). This seems like a narrow framing of the results, which found at least some predictors in each of the broad domains were associated with lack of interest in sex. I think it would be good to focus instead on the breadth and variety of predictors, particularly those with substantial effect sizes, interpreting how these findings can inform research and practice on lack of sexual interest.

REVIEWER	Ashley E Thompson University of Wisconsin Oshkosh, United States
REVIEW RETURNED	24-Apr-2017

GENERAL COMMENTS	The paper entitled “Factors associated with reporting lacking interest in sex and their interaction with gender: Findings from the third British National Survey of Sexual Attitudes and Lifestyles” was incredibly informative and is a great contribution to the literature. Despite the wealth of information included, some revisions could substantially improve the already educational manuscript. I have outlined some of these potential revisions below. 1. Although it is important to keep manuscripts as succinct as possible, I encourage the authors to provide more background information about some of the factors that were assessed regarding their impact on low interest in sexual behavior. For example little information was given about why sexual attitudinal variables and the type of sexual behavior would be related to low sexual desire. Providing more of a background can help inform the reader as to why these variables were selected for further investigation. Furthermore, although this paper was largely exploratory, there was definitely room for hypotheses. Because research projects that fail to include a theoretical approach/background often lack scientific merit, I encourage the authors to identify hypotheses based on the previous research outlined in the introduction. 2. In order to assess the validity of the study and to encourage replications, more information should be provided in the method section. In particular, the number of men and women who participated should be included in the participant section (despite being included in the abstract). a. I was also a bit confused about what the CAPI and the CASI procedures entailed. Seeing as though it is likely that other readers may also be unfamiliar with these procedures, it would be advantageous for the authors to expand on these techniques, potentially in a procedure section (this would provide a great opportunity to flesh out every step in detail). b. Although I recognize that I can refer to other articles with which some of the methods are outlined, I think more information should be provided for your variables of study, particularly your outcome variables/measures. For example, how many items were selected to represent your outcome measure? What was the response format for each? Were metrics of internal consistency computed (Cronbach’s alphas)? 3. The authors should also attempt interpreting their findings in the results section. By this, I do not mean that they should explain or discuss, but to provide more detail. An example would be in the final results paragraph (beginning in line 18 on page 10). The authors mentioned that there was an association between lacking enjoyment and lacking interest, but the direction of the association is unclear (although I expected it is assumed). An interpretation as to whether this relationship was positive or negative will help the reader better understand the nature of the trends. Minor Concerns 4. I encourage the authors to be careful with their terminology. In the introduction, the authors seem to use “women” and “females” as well as “men” and “males” interchangeably. If this is a paper on
--

	gender differences, then the terms “men” and “women” should be used. 5. The authors should also carefully proof the paper for formatting inconsistencies. For example, in line 24 on page 10, the font changes. 6. At times the wording was a bit off. For example, in line 15 on page 6, the authors mention that “fuller details of the survey methodology Are published elsewhere.” However, I do not believe “fuller” is a word. Perhaps the authors could revise to “An extensive review of the survey methodology...”
--	---

VERSION 1 – AUTHOR RESPONSE

Reviewer: 1

Reviewer Name: Miriam K. Forbes, PhD

Institution and Country: University of Minnesota, USA Please state any competing interests: None declared.

This paper uses the NATSAL-3 data to explicate the sociodemographic; mental and physical health; sexual health, behaviours and attitudes; relationship; and lifestyle factors associated with a lack of interest in sex that has lasted for three months or more in the past year. The NATSAL-3 data are ideal for these analyses, particularly with the use of methodological variables that account for the complex survey design. The aim of painting a detailed picture of the correlates of lacking sexual interest is also important (although I think the authors need to make this stronger case in-text). I would suggest that the manuscript could be strengthened by some changes in the statistical analyses and the reporting of results in particular, as described below. I have also listed some other questions, comments, and suggestions below that may help to strengthen the manuscript.

Response: Thank you for the positive comment. As stated in our response to point 1 below, we have provided a stronger rationale for why it is important to understand the correlates of lacking sexual interest.

Introduction

1. The intro is brief and to the point, but I think that it needs a paragraph to develop a rationale for why it is important to understand the correlates of lacking sexual interest.

Response: We have added some additional text in the Introduction (p. 5) on why it is important to understand the correlates of lacking sexual interest.

Method

2. These data are ideal for addressing the aims of the study.

Response: Thank you – we agree!

3. Why were participants required to have a sexual partner in the past year to be included in this study?

Response: The relevant questions were only asked of those with 1+ sexual partner in the past year (but note that we did not restrict our analyses to men and women in a relationship, only to those who reported having had a sexual partner in the last year). We acknowledge that this focus excludes those who had not had sex in the past year because of lack of interest in sex and include this as a limitation in the Discussion.

4. The associations between lacking interest in sex and other sexual function problems seem tangential to the main aims of the study.

Response: We appreciate the reviewer's comment here, as we did not include any specific research question related to these analyses. To address this concern, we have added an additional research question (Introduction, page 5).

Results and Discussion

5. Overall, I think the results need a little more attention to detail. For example, some of the significant results were not reported in-text:

a. the index of multiple deprivation was a significant predictor for women
b. employment status was also significant for women
c. number of comorbid health conditions was a significant predictor for men and women
d. taking drugs to assist sexual performance was a predictor for men
e. relationship status was differentially related to lacking interest in sex for women (i.e., all categories had lower AORs than living with partner) vs men
f. non-volitional sex was a significant predictor for men and women.

Response: We have added additional text to report on all but one of the significant results noted above. We did not include text on the Index of Multiple Deprivation variable because the results were not strong and were also not maintained in the sensitivity analysis.

6. There are also some examples where multiple response categories were conflated in-text:

a. "at least three sexual acts" is 3-4 sexual acts specifically, with separate results (albeit the same pattern) for 5+
b. "women with two or more partners" (vs. 1) AOR is for 3+ partners.

Response: Thank you for spotting these errors – both of these sentences have been corrected.

7. There are a few times where it needs to be specified that an effect was gender-specific:

a. pregnancy and children under 5
b. sexual competence at first sex.

Response: We have revised the text to indicate that these associations were gender-specific.

These three points (5-7) are obviously nitpicking --particularly given the number of results included in Table 1 and reported in-text-- but since the aim of the paper is to be descriptive, I think particularly careful attention to detail is warranted.

8. I also think it is important that the significance levels are adjusted for the multiple comparisons. It's good that the authors currently interpret the effect sizes (AORs), rather than purely the significance, but given the large number of predictors and the large sample size, a p-value of .05 will likely include a substantial number of false positive results. Similarly, the reported CIs should be proportionately widened.

Response: In common with many epidemiological studies we have tested many associations within this study. However, we do not feel that formal correction of p-values would be appropriate firstly because there is no universally accepted method to do this, but mainly because we think it is more appropriate to interpret the results holistically and to be suitably cautious in the interpretation of p-values. We would prefer to keep our significance level at 0.05 but have added a sentence into the discussion about exercising caution when concluding associations where $0.01 < p$

9. It might also be interesting (although not necessary) to look at the unique predictions of the significant correlates - e.g., to answer questions like does relationship status predict lack of interest in sex over and above relationship duration?

Response: While we agree with the reviewer that these types of questions would indeed be interesting, we think that further analyses along these lines should be hypothesis-driven. In our view, we don't think that the previous research in this area does suggest any specific hypotheses that should be tested. However, if the Editor would prefer that we do conduct some further analyses along these lines, we would be happy to do so.

10. There is some richness in the sensitivity analyses (Table 2) that may make the discussion more

robust - for example, there are significant interactions for gender and the depression variables (stronger effects for men); women's masturbation is no longer associated with lack of interest in sex (but men's is); pregnancy is no longer a significant predictor, etc. In the context of considering lacking interest in sex in general vs. a distressing lack of interest in sex, these differences in the results are potentially interesting and worth spending time on.

Response: As suggested, we have added some additional text to the Discussion to discuss some of the findings in the sensitivity analyses. We have been selective about how much text to add because we did not want to add significantly to the length of the paper.

11. If space is an issue (and it's an option) it might be worth combining the results and discussion to avoid the need to re-state the results in the discussion. Alternatively, referring readers to the table without spending time interpreting each result in-text (in the results section) might free up space to go into more detail for some points in the discussion.

Response: We considered doing this but decided against combining the Results and Discussion sections, as our word count is still within the maximum recommended by the journal. We were also encouraged to add more detail in the Results section (by both this reviewer and reviewer 2), which we have now done.

12. The implications section opens with "Our findings underscore the importance of the relational context in understanding low sexual interest in both men and women" (p. 14). This seems like a narrow framing of the results, which found at least some predictors in each of the broad domains were associated with lack of interest in sex. I think it would be good to focus instead on the breadth and variety of predictors, particularly those with substantial effect sizes, interpreting how these findings can inform research and practice on lack of sexual interest.

Response: We agree with the reviewer that the beginning of this section focused on a narrow framing of the results and we have revised the implications section to include a broader focus on our findings (pp. 11-12)

Reviewer: 2

Reviewer Name: Ashley E Thompson

Institution and Country: University of Wisconsin Oshkosh, United States

Please state any competing interests: None declared

The paper entitled "Factors associated with reporting lacking interest in sex and their interaction with gender: Findings from the third British National Survey of Sexual Attitudes and Lifestyles" was incredibly informative and is a great contribution to the literature. Despite the wealth of information included, some revisions could substantially improve the already educational manuscript. I have outlined some of these potential revisions below.

Response: Thank you for this positive feedback.

1. Although it is important to keep manuscripts as succinct as possible, I encourage the authors to provide more background information about some of the factors that were assessed regarding their impact on low interest in sexual behavior. For example, little information was given about why sexual attitudinal variables and the type of sexual behavior would be related to low sexual desire. Providing more of a background can help inform the reader as to why these variables were selected for further investigation. Furthermore, although this paper was largely exploratory, there was definitely room for hypotheses. Because research projects that fail to include a theoretical approach/background often lack scientific merit, I encourage the authors to identify hypotheses based on the previous research outlined in the introduction.

Response: We have added some additional background information about why some of the factors we assessed might be related to low sexual desire. However, as we noted in the Introduction, few studies have examined possible links between lacking interest in sex and sexual attitudes and behaviour, such that our paper's main focus is on hypothesis-generating based on data

representative of the population (vs. clinic samples).

2. In order to assess the validity of the study and to encourage replications, more information should be provided in the method section. In particular, the number of men and women who participated should be included in the participant section (despite being included in the abstract).

Response: We have added the number of men and women who participated to the Participants section.

a. I was also a bit confused about what the CAPI and the CASI procedures entailed. Seeing as though it is likely that other readers may also be unfamiliar with these procedures, it would be advantageous for the authors to expand on these techniques, potentially in a procedure section (this would provide a great opportunity to flesh out every step in detail).

Response: We have reorganized the text in the Method section and hope that the procedure is clearer now. We have also briefly expanded on use of the CAPI and CASI techniques (p. 7).

b. Although I recognize that I can refer to other articles with which some of the methods are outlined, I think more information should be provided for your variables of study, particularly your outcome variables/measures. For example, how many items were selected to represent your outcome measure? What was the response format for each? Were metrics of internal consistency computed (Cronbach's alphas)?

Response: We have provided additional information about the outcome measures, including the questions and the response format for each.

3. The authors should also attempt interpreting their findings in the results section. By this, I do not mean that they should explain or discuss, but to provide more detail. An example would be in the final results paragraph (beginning in line 18 on page 10). The authors mentioned that there was an association between lacking enjoyment and lacking interest, but the direction of the association is unclear (although I expected it is assumed). An interpretation as to whether this relationship was positive or negative will help the reader better understand the nature of the trends.

Response: Reviewer 1 also suggested providing more detail in the Results section (point 5 above) and we have done this. We also clarified in the text that the association between lacking enjoyment and lacking interest was positive.

Minor Concerns

4. I encourage the authors to be careful with their terminology. In the introduction, the authors seem to use "women" and "females" as well as "men" and "males" interchangeably. If this is a paper on gender differences, then the terms "men" and "women" should be used.

Response: We agree that the terms "men" and "women" should be used, but when we did a search for the terms "male" and "females" the only instances where we used these were as adjectives e.g., "male disorders", etc. As the use of "female" and "male" as adjectives seems appropriate, we elected not to change these.

5. The authors should also carefully proof the paper for formatting inconsistencies. For example, in line 24 on page 10, the font changes.

Response: We have checked for formatting inconsistencies and corrected the text above.

At times the wording was a bit off. For example, in line 15 on page 6, the authors mention that "fuller details of the survey methodology Are published elsewhere." However, I do not believe "fuller" is a

word. Perhaps the authors could revise to “An extensive review of the survey methodology...”

Response: We have revised this text and replaced “fuller” with “a more extensive.”

VERSION 2 – REVIEW

REVIEWER	Miriam K. Forbes, PhD University of Minnesota, USA
REVIEW RETURNED	20-Jun-2017

GENERAL COMMENTS	The authors have addressed the points I raised in my initial review, and I appreciate their considered responses. As before, I believe this paper makes an important contribution to the literature. I have only a few minor suggestions for this revision: 1. I would suggest revising the last sentence of Participants and Procedure (p. 6) to explain why this analytic sample was selected (moving up content from the next section), for example: "Only respondents who reported at least one sexual partner in the past year (4839 men and 6669 women) were asked whether they had lacked interest in sex for a period of three months or longer (see below). These participants are the focus of the current analyses."2. The inclusion of model fit indices in the first sentence of Outcome Measures (p. 7) was a bit confusing—they don't have clear relevance and could probably be excluded.3. The authors have chosen not to adjust for multiple comparisons in their analyses. Focusing on the AORs is indeed better than focusing on statistical significance, but I think it is important to extend the limitations to emphasise that these are exploratory and descriptive analyses of zero-order relationships. As such, the smaller effect sizes may not replicate and may not hold in multivariate analyses (i.e., net of one another)—i.e., rather than the current closing sentence of the limitations advising caution in interpreting effects with significance $.01 < p < .05$.
---

VERSION 2 – AUTHOR RESPONSE

1. I would suggest revising the last sentence of Participants and Procedure (p. 6) to explain why this analytic sample was selected (moving up content from the next section), for example: "Only respondents who reported at least one sexual partner in the past year (4839 men and 6669 women) were asked whether they had lacked interest in sex for a period of three months or longer (see below). These participants are the focus of the current analyses."

Response: We have made this change.

2. The inclusion of model fit indices in the first sentence of Outcome Measures (p. 7) was a bit confusing—they don't have clear relevance and could probably be excluded.

Response: We have deleted this part of the sentence.

3. The authors have chosen not to adjust for multiple comparisons in their analyses. Focusing on the

AORs is indeed better than focusing on statistical significance, but I think it is important to extend the limitations to emphasise that these are exploratory and descriptive analyses of zero-order relationships. As such, the smaller effect sizes may not replicate and may not hold in multivariate analyses (i.e., net of one another)—i.e., rather than the current closing sentence of the limitations advising caution in interpreting effects with significance